# Equivariant Flow Matching for Point Cloud Assembly

## Abstract

The goal of point cloud assembly is to reconstruct a complete 3D shape by aligning multiple point cloud pieces. This work presents a novel equivariant solver for assembly tasks based on flow matching models. We first theoretically show that the key to learning equivariant distributions via flow matching is to learn related vector fields. Based on this result, we propose an assembly model, called equivariant diffusion assembly (Eda), which learns related vector fields conditioned on the input pieces. We further construct an equivariant path for Eda, which guarantees high data efficiency of the training process. Our numerical results show that Eda is highly competitive on practical datasets, and it can even handle the challenging situation where the input pieces are non-overlapped.

## 1 Introduction

Point cloud (PC) assembly is a classic machine learning task which seeks to reconstruct 3D shapes by aligning multiple point cloud pieces. This task has been intensively studied for decades and has various applications such as scene reconstruction (Zeng et al., 2017), robotic manipulation (Ryu et al., 2024), cultural relics reassembly (Wang et al., 2021) and protein designing (Watson et al., 2023). A key challenge in this task is to correctly align PC pieces with small or no overlap region, *i.e.*, when the correspondences between pieces are lacking.

To address this challenge, some recent methods (Ryu et al., 2024; Wang and Jörnsten, 2024) utilized equivariance priors for pair-wise assembly tasks, *i.e.*, the assembly of two pieces. In contrast to most of the state-of-the-art methods (Qin et al., 2022; Zhang, 1994) which align PC pieces based on the inferred correspondence, these equivariant methods are correspondence-free, and they are guided by the equivariance law underlying the assembly task. As a result, these methods are able to assemble PCs without correspondence, and they enjoy high data efficiency and promising accuracy. However, the extension of these works to multi-piece assembly tasks remains largely unexplored.

In this work, we develop an equivariant method for multi-piece assembly based on flow matching (Lipman et al., 2023). Our main theoretical finding is that to learn an equivariant distribution via flow matching, one only needs to ensure that the initial noise is invariant and the vector field is related (Thm. 4.2). In other words, instead of directly handling the $SE(3)^N$-equivariance for $N$-piece assembly tasks, which can be computationally expensive, we only need to handle the related vector fields on $SE(3)^N$, which is efficient and easy to construct. Based on this result, we present a novel assembly model called equivariant diffusion assembly (Eda), which uses invariant noise and predicts related vector fields by construction. Eda is correspondence-free and is guaranteed to be equivariant by our theory. Furthermore, we construct a short and equivariant path for the training of Eda, which guarantees high data efficiency of the training process. When Eda is trained, an assembly solution can be sampled by numerical integration, *e.g.*, the Runge-Kutta method, starting from a random noise. All proofs can be found in Appx. F. A brief walk-through of our theory using a toy example with minimal terminologies is provided in Appx. C

The contributions of this work are summarized as follows:

- We present an equivariant flow matching framework for multi-piece assembly tasks. Our theory reduces the task of constructing equivariant conditional distributions to the task of constructing related vector fields, thus it provides a feasible way to define equivariant flow matching models.

- Based on the theoretical result, we present a simple and efficient multi-piece PC assembly model, called equivariant diffusion assembly (Eda), which is correspondence-free and is guaranteed to be equivariant. We further construct an equivariant path for the training of Eda, which guarantees high data efficiency.

- We numerically show that Eda produces highly accurate results on the challenging 3DMatch and BB datasets, and it can even handle non-overlapped pieces.

## 2 RELATED WORK

Our proposed method is based on flow matching (Lipman et al., 2023), which is one of the state-of-the-art diffusion models for image generation tasks (Esser et al., 2024). Some applications on manifolds have also been investigated (Chen and Lipman, 2024; Yim et al., 2023). Our model has two distinguishing features compared to existing methods: it learns conditional distributions instead of marginal distributions, and it explicitly incorporates equivariance priors.

The PC assembly task studied in this work is related to various tasks in the literature, such as PC registration (Qin et al., 2022; Yu et al., 2023), robotic manipulation (Ryu et al., 2024; 2023) and fragment reassembly (Wu et al., 2023a). All these tasks aim to align the input PC pieces, but they are different in settings such as the number of pieces, deterministic or probabilistic, and whether the PCs are overlapped. More details can be found in Appx. B. In this work, we consider the most general setting: we aim to align multiple pieces of non-overlapped PCs in a probabilistic way.

Recently, diffusion-based methods have been proposed for assembly tasks (Chen et al., 2025; Jiang et al., 2023; Wu et al., 2023b; Li et al., 2025; Ryu et al., 2024; Scarpellini et al., 2024; Xu et al., 2024). However, most of these works ignore the manifold structure or the equivariance priors of the task. One notable exception is Ryu et al. (2024), which developed an equivariant diffusion method for robotic manipulation, *i.e.*, pair-wise assembly tasks. Compared to Ryu et al. (2024), our method is conceptually simpler because it does not require Brownian diffusion on $SO(3)$ whose kernel is computationally intractable, and it solves the more general multi-piece problem. On the other hand, the invariant flow theory has been studied in Köhler et al. (2020), which can be regarded as a special case of our theory as discussed in Appx. F.1. Furthermore, the optimal-transport-based method was explored for invariant flow (Song et al., 2023; Klein et al., 2023).

Another branch of related work is equivariant neural networks. Due to their ability to incorporate geometric priors, this type of networks has been widely used for processing 3D graph data such as PCs and molecules. In particular, E3NN (Geiger and Smidt, 2022) is a well-known equivariant network based on the tensor product of the input and the edge feature. An acceleration technique for E3NN was recently proposed (Passaro and Zitnick, 2023). On the other hand, the equivariant attention layer was studied in Fuchs et al. (2020); Liao and Smidt (2023); Liao et al. (2024). Our work is related to this line of approach, because our diffusion network can be seen as an equivariant network with an additional time parameter.

## 3 PRELIMINARIES

This section introduces the major tools used in this work. We first define the equivariances in Sec. 3.1, then we briefly recall the flow matching model in Sec. 3.2.

### 3.1 EQUIVARIANCES OF PC ASSEMBLY

Consider the action $G = \prod_{i=1}^{N} SE(3)$ on a set of $N$ ($N \geq 2$) PCs $X = \{X_1, \ldots, X_N\}$, where $SE(3)$ is the 3D rigid transformation group, $\prod$ is the direct product, and $X_i$ is the i-th PC piece in 3D space. We define the action of $\boldsymbol{g} = (g_1, \ldots, g_N) \in G$ on $X$ as $\boldsymbol{g}X = \{g_i X_i\}_{i=1}^{N}$, *i.e.*, each PC $X_i$ is rigidly transformed by the corresponding $g_i$. For the rotation subgroup $SO(3)^N$, the action of $\boldsymbol{r} = (r_1, \ldots, r_N) \in SO(3)^N$ on $X$ is $\boldsymbol{r}X = \{r_i X_i\}_{i=1}^{N}$. For $SO(3) \subseteq G$, we denote $r = (r, \ldots, r) \in SO(3)$ for simplicity, and the action of $r$ on $X$ is written as $rX = \{rX_i\}_{i=1}^{N}$.

We also consider the permutations of $X$. Let $S_N$ be the permutation group of $N$, the action of $\sigma \in S_N$ on $X$ is $\sigma X = \{X_{\sigma(i)}\}_{i=1}^{N}$, and the action on $\boldsymbol{g}$ is $\sigma\boldsymbol{g} = (g_{\sigma(1)}, \ldots, g_{\sigma(N)})$. For group multiplication,

we denote $\mathcal{R}_{(\cdot)}$ the right multiplication and $\mathcal{L}_{(\cdot)}$ the left multiplication, *i.e.*, $(\mathcal{R}_{\boldsymbol{r}})\boldsymbol{r}' = \boldsymbol{r}'\boldsymbol{r}$, and $(\mathcal{L}_{\boldsymbol{r}})\boldsymbol{r}' = \boldsymbol{r}\boldsymbol{r}'$ for $\boldsymbol{r}, \boldsymbol{r}' \in SO(3)^N$.

In our setting, for the given input $X$, the solution to the assembly task is a conditional distribution $P_X \in \mu(G)$, where $\mu(G)$ is the set of probability distribution on $G$. We study the following three equivariances of $P_X$ in this work:

**Definition 3.1.** Let $P_X \in \mu(G)$ be a probability distribution on $G = SE(3)^N$ conditioned on $X$, and let $(\cdot)_\#$ be the pushforward of measures.

- $P_X$ is $SO(3)^N$-equivariant if $(\mathcal{R}_{\boldsymbol{r}^{-1}})_\# P_X = P_{\boldsymbol{r}X}$ for $\boldsymbol{r} \in SO(3)^N$.

- $P_X$ is permutation-equivariant if $\sigma_\# P_X = P_{\sigma X}$ for $\sigma \in S_N$.

- $P_X$ is $SO(3)$-invariant if $(\mathcal{L}_r)_\# P_X = P_X$ for $r \in SO(3)$.

As an example, we explicitly show the equivariance in Def. 3.1 for a two-piece deterministic problem.

**Example 3.2.** Assume that a solution for point clouds $(X_1, X_2)$ is $(r_1, r_2)$, meaning $r_1 X_1$ and $r_2 X_2$ are assembled, then

- $SO(3)^2$-equivariance: a solution for $(r_3 X_1, r_4 X_2)$ is $(r_1 r_3^{-1}, r_2 r_4^{-1})$;

- Permutation-equivariance: a solution for $(X_2, X_1)$ is $(r_2, r_1)$;

- $SO(3)$-invariance: another solution for $(X_1, X_2)$ is $(rr_1, rr_2)$.

More discussions on the definition of equivariances can be found in Appx. D

We finally recall the definition of $SO(3)$-equivariant networks, which will be the main computational tool of this work. We call $F^l \in \mathbb{R}^{2l+1}$ a degree-$l$ $SO(3)$-equivariant feature if the action of $r \in SO(3)$ on $F^l$ is the matrix-vector production: $rF^l = R^l F^l$, where $R^l \in \mathbb{R}^{(2l+1)\times(2l+1)}$ is the degree-$l$ Wigner-D matrix of $r$. We call a network $w$ $SO(3)$-equivariant if it maintains the equivariance from the input to the output: $w(rX) = rw(X)$, where $w(X)$ is a $SO(3)$-equivariant feature. More detailed introduction of equivariances and the underlying representation theory can be found in Cesa et al. (2022).

## 3.2 VECTOR FIELDS AND FLOW MATCHING

To sample from a data distribution $P_1 \in \mu(M)$, where $M$ is a smooth manifold (we only consider $M = G$ in this work), the flow matching (Lipman et al., 2023) approach constructs a time-dependent diffeomorphism $\phi_\tau : M \to M$ satisfying $(\phi_0)_\# P_0 = P_0$ and $(\phi_1)_\# P_0 = P_1$, where $P_0 \in \mu(M)$ is a fixed noise distribution, and $\tau \in [0, 1]$ is the time parameter. Then the sample of $P_1$ can be represented as $\phi_1(g)$ where $g$ is sampled from $P_0$.

Formally, $\phi_\tau$ is defined as a flow, *i.e.*, an integral curve, generated by a time-dependent vector field $v_\tau : M \to TM$, where $TM$ is the tangent bundle of $M$:

$$\frac{\partial}{\partial \tau} \phi_\tau(\boldsymbol{g}) = v_\tau(\phi_\tau(\boldsymbol{g})),$$
$$\phi_0(\boldsymbol{g}) = \boldsymbol{g}, \quad \forall \boldsymbol{g} \in M. \tag{1}$$

According to Lipman et al. (2023), an efficient way to construct $v_\tau$ is to define a path $h_\tau$ connecting $P_0$ to $P_1$. Specifically, let $\boldsymbol{g}_0$ and $\boldsymbol{g}_1$ be samples from $P_0$ and $P_1$ respectively, and $h_0 = \boldsymbol{g}_0$ and $h_1 = \boldsymbol{g}_1$. $v_\tau$ can be constructed as the solution to the following problem:

$$\min_v \mathbb{E}_{\tau, \boldsymbol{g}_0 \sim P_0, \boldsymbol{g}_1 \sim P_1} ||v_\tau(h_\tau) - \frac{\partial}{\partial \tau} h_\tau||_F^2. \tag{2}$$

When $v$ is learned using (2), we can obtain a sample from $P_1$ by first sampling a noise $\boldsymbol{g}_0$ from $P_0$ and then taking the integral of (1).

In this work, we consider a family of vector fields, flows and paths conditioned on the given PC, and we use the pushforward operator on vector fields to study their relatedness (Tu, 2011). Formally, let $F : M \to M$ be a diffeomorphism, $v$ and $w$ be vector fields on $M$. $w$ is $F$-related to $v$ if $w(F(\boldsymbol{g})) = F_{*,\boldsymbol{g}} v(\boldsymbol{g})$ for all $\boldsymbol{g} \in M$, where $F_{*,\boldsymbol{g}}$ is the differential of $F$ at $\boldsymbol{g}$. Note that we denote $v_X$, $\phi_X$ and $h_X$ the vector field, flow and path conditioned on PC $X$ respectively.

*Remark* 3.3. For readers that are not familiar with this definition, relatedness can be simply regarded as a transformation, so the above definition simply means $w$ is the transformation of $v$ by $F$. More details can be found in Sec.14.6 in the text book Tu (2011).

# 4 METHOD

In this section, we provide the details of the proposed Eda model. First, the PC assembly problem is formulated in Sec. 4.1. Then, we parametrize related vector fields in Sec. 4.2. The training and sampling procedures are finally described in Sec. 4.3 and Sec. 4.4 respectively.

## 4.1 PROBLEM FORMULATION

Given a set $X$ containing $N$ PC pieces, *i.e.*, $X = \{X_i\}_{i=1}^N$ where $X_i$ is the $i$-th piece, the goal of assembly is to learn a distribution $P_X \in \mu(G)$, *i.e.*, for any sample $g$ of $P_X$, $gX$ should be the aligned complete shape. We assume that $P_X$ has the following equivariances:

**Assumption 4.1.** $P_X$ is $SO(3)^N$-equivariant, permutation-equivariant and $SO(3)$-invariant.

We seek to approximate $P_X$ using flow matching. To avoid translation ambiguity, we also assume that, without loss of generality, the aligned PCs $gX$ and each input piece $X_i$ are centered, *i.e.*, $\sum_i \mathbf{m}(g_iX_i) = 0$, and $\mathbf{m}(X_i) = 0$ for all $i$, where $\mathbf{m}(\cdot)$ is the mean vector.

## 4.2 EQUIVARIANT FLOW

The major challenge in our task is to ensure the equivariance of the learned distribution, because a direct implementation of flow matching (1) generally does not guarantee any equivariance. To address this challenge, we utilize the following theorem, which claims that when the noise distribution $P_0$ is invariant and vector fields $v_X$ are related, the pushforward distribution $(\phi_X)\#P_0$ is guaranteed to be equivariant.

**Theorem 4.2.** *Let $G$ be a smooth manifold, $F : G \to G$ be a diffeomorphism, and $P \in \mu(G)$. If vector field $v_X \in TG$ is $F$-related to vector field $v_Y \in TG$, then*

$$F_\#P_X = P_Y, \tag{3}$$

*where $P_X = (\phi_X)_\#P_0$, $P_Y = (\phi_Y)_\#(F_\#P_0)$. Here $\phi_X, \phi_Y : G \to G$ are generated by $v_X$ and $v_Y$ respectively.*

Specifically, Thm. 4.2 provides a concrete way to construct the three equivariances required by Assumption 4.1 as follow.

**Assumption 4.3** (Invariant noise). $P_0$ is $SO(3)^N$-invariant, permutation-invariant and $SO(3)$-invariant, *i.e.*, $(\mathcal{R}_{\mathbf{r}^{-1}})_\#P_0 = P_0$, $\sigma_\#P_0 = P_0$ and $P_0 = (\mathcal{L}_r)_\#P_0$ for $\mathbf{r} \in SO(3)^N$, $\sigma \in S_N$ and $r \in SO(3)$.

**Corollary 4.4.** *Under assumption 4.3,*

- *if $v_X$ is $\mathcal{R}_{\mathbf{r}^{-1}}$-related to $v_{\mathbf{r}X}$, then $(\mathcal{R}_{\mathbf{r}^{-1}})_\#P_X = P_{\mathbf{r}X}$, where $P_X = (\phi_X)_\#P_0$ and $P_{\mathbf{r}X} = (\phi_{\mathbf{r}X})_\#P_0$. Here $\phi_X, \phi_{\mathbf{r}X} : G \to G$ are generated by $v_X$ and $v_{\mathbf{r}X}$ respectively.*

- *if $v_X$ is $\sigma$-related to $v_{\sigma X}$, then $\sigma_\#P_X = P_{\sigma X}$, where $P_X = (\phi_X)_\#P_0$ and $P_{\sigma X} = (\phi_{\sigma X})_\#P_0$. Here $\phi_X, \phi_{\sigma X} : G \to G$ are generated by $v_X$ and $v_{\sigma X}$ respectively.*

- *if $v_X$ is $\mathcal{L}_r$-invariant, i.e., $v_X$ is $\mathcal{L}_r$-related to $v_X$, then $(\mathcal{L}_r)_\#P_X = P_X$, where $P_X = (\phi_X)_\#P_0$.*

According to Cor. 4.4, if the vector fields $v_X$ are related, then the solution $P_X$ is guaranteed to be equivariant. Therefore, the problem is reduced to constructing related vector fields. We start by constructing $(\mathcal{R}_{\mathbf{g}^{-1}})$-related vector fields, which are $(\mathcal{R}_{\mathbf{r}^{-1}})$-related by definition, where $\mathbf{g} \in SE(3)^N$ and $\mathbf{r} \in SO(3)^N$. Specifically, we have the following proposition:

**Proposition 4.5.** $v_X$ is $\mathcal{R}_{\mathbf{g}^{-1}}$-related to $v_{\mathbf{g}X}$ if and only if $v_X(\mathbf{g}) = v_{\mathbf{g}X}(e)\mathbf{g}$ for all $\mathbf{g} \in SE(3)^N$.

Prop. 4.5 suggests that for $(\mathcal{R}_{\boldsymbol{g}^{-1}})$-related vector fields $v_X$, $v_X(\boldsymbol{g})$ is fully determined by the value of $v_{\boldsymbol{g}X}$ at the identity element $e$. Therefore, to parametrize $v_X$, we only need to parametrize $v_{\boldsymbol{g}X}$ at one single point $e$. Specifically, let $f$ be a neural network parametrizing $v_X(e)$ for input $X$, *i.e.*, $f(X) = v_X(e)$, $v_X$ can then be written as

$$v_X(\boldsymbol{g}) = f(\boldsymbol{g}X)\boldsymbol{g}. \tag{4}$$

Here, $f(X) \in \mathfrak{se}(3)^N$ takes the form of

$$f(X) = \bigoplus_{i=1}^{N} f_i(X) \quad \text{where} \quad f_i(X) = \begin{pmatrix} w^i_\times(X) & t^i(X) \\ 0 & 0 \end{pmatrix} \in \mathfrak{se}(3) \subseteq \mathbb{R}^{4\times 4}. \tag{5}$$

The rotation component $w^i_\times(X) \in \mathbb{R}^{3\times3}$ is a skew matrix with elements in the vector $w^i(X) \in \mathbb{R}^3$, and $t^i(X) \in \mathbb{R}^3$ is the translation component. For simplicity, we omit the superscript $i$ when the context is clear.

Now we proceed to the other two types of relatedness of $v_X$. According to the following proposition, when $v_X$ is written as (4), these two relatedness of $v_X$ can be guaranteed if the network $f$ is equivariant.

**Proposition 4.6.** *For $v_X$ defined in (4),*

- *if $f$ is permutation-equivariant, i.e., $f(\sigma X) = \sigma f(X)$ for $\sigma \in S_N$ and PCs $X$, then $v_X$ is $\sigma$-related to $v_{\sigma X}$.*

- *if $f$ is SO(3)-equivariant, i.e., $w(rX) = rw(X)$ and $t(rX) = rt(X)$ for $r \in SO(3)$ and PCs $X$, then $v_X$ is $\mathcal{L}_r$-invariant.*

Finally, we define $P_0 = (U_{SO(3)} \otimes \mathcal{N}(0, \omega I))^N$, where $U_{SO(3)}$ is the uniform distribution on $SO(3)$, $\mathcal{N}$ is the normal distribution on $\mathbb{R}^3$ with mean zero and isotropic variance $\omega \in \mathbb{R}_+$, and $\otimes$ represents the independent coupling. It is straightforward to verify that $P_0$ indeed satisfies assumption 4.3.

In summary, with $P_0$ and $v$ constructed above, the learned distribution is guaranteed to be $SO(3)^N$-equivariance, permutation-equivariance and $SO(3)$-invariance.

## 4.3 TRAINING

To learn the vector field $v_X$ (4) using flow matching (2), we now need to define $h_X$, and the sampling strategy of $\tau$, $\boldsymbol{g}_0$ and $\boldsymbol{g}_1$. A canonical choice (Chen and Lipman, 2024) is $\bar{h}(\tau) = \boldsymbol{g}_0 \exp(\tau \log(\boldsymbol{g}_0^{-1}\boldsymbol{g}_1))$, where $\boldsymbol{g}_0$ and $\boldsymbol{g}_1$ are sampled independently, and $\tau$ is sampled from a predefined distribution, *e.g.*, the uniform distribution $U_{[0,1]}$. However, this definition of $h$, $\boldsymbol{g}_0$ and $\boldsymbol{g}_1$ does not utilize any equivariance property of $v_X$, thus it does not guarantee a high data efficiency.

To address this issue, we construct a "short" and equivariant $h_X$ in the following two steps. First, we independently sample $\boldsymbol{g}_0$ from $P_0$ and $\tilde{\boldsymbol{g}}_1$ from $P_X$, and obtain $\boldsymbol{g}_1 = r^*\tilde{\boldsymbol{g}}_1$, where $r^* \in SO(3)$ is a rotation correction of $\tilde{\boldsymbol{g}}_1$:

$$r^* = \underset{r \in SO(3)}{\arg\min} ||r\tilde{\boldsymbol{g}}_1 - \boldsymbol{g}_0||_F^2. \tag{6}$$

Then, we define $h_X$ as

$$h_X(\tau) = \exp(\tau \log(\boldsymbol{g}_1\boldsymbol{g}_0^{-1}))\boldsymbol{g}_0. \tag{7}$$

We call $h_X$ (7) a path generated by $\boldsymbol{g}_0$ and $\tilde{\boldsymbol{g}}_1$. A similar rotation correction in the Euclidean space was studied in Song et al. (2023); Klein et al. (2023). Note that $h_X$ (7) is a well-defined path connecting $\boldsymbol{g}_0$ to $\boldsymbol{g}_1$, because $h_X(0) = \boldsymbol{g}_0$ and $h_X(1) = \boldsymbol{g}_1$, and $\boldsymbol{g}_1$ follows $P_X$ (Prop. F.5).

The advantages of $h_X$ (7) are twofold. First, instead of connecting a noise $\boldsymbol{g}_0$ to an independent data sample $\tilde{\boldsymbol{g}}_1$, $h_X$ connects $\boldsymbol{g}_0$ to a modified sample $\boldsymbol{g}_1$ where the redundant rotation component is removed, thus it is easier to learn. Second, the velocity fields of $h_X$ enjoy the same relatedness as $v_X$ (4), which leads to high data efficiency. Formally, we have the following observation.

**Proposition 4.7** (Data efficiency). *Under assumption 4.3, 4.1, and F.4, we further assume that $v_X$ satisfies the relatedness property required in Cor. 4.4, i.e., $v_X$ is $\mathcal{R}_{\boldsymbol{r}^{-1}}$-related to $v_{\boldsymbol{r}X}$, $v_X$ is $\sigma$-related to $v_{\sigma X}$, and $v_X$ is $\mathcal{L}_r$-invariant. Denote $L(X) = \mathbb{E}_{\tau, \boldsymbol{g}_0 \sim P_0, \tilde{\boldsymbol{g}}_1 \sim P_X}||v_X(h_X(\tau)) - \frac{\partial}{\partial \tau}h_X(\tau)||_F^2$ the training loss (2) of PC $X$, where $h_X$ is generated by $\boldsymbol{g}_0$ and $\tilde{\boldsymbol{g}}_1$ as defined in (7). Then*

- $L(X) = L(\boldsymbol{r}X)$ *for* $\boldsymbol{r} \in SO(3)^N$.

- $L(X) = L(\sigma X)$ *for* $\sigma \in S_N$.

- $L(X) = \hat{L}(X)$, *where* $\hat{L}(X) = \mathbb{E}_{\tau, \boldsymbol{g}_0' \sim P_0, \tilde{\boldsymbol{g}}_1' \sim (\mathcal{L}_r)_{\#} P_X} ||v_X(h_X(\tau)) - \frac{\partial}{\partial \tau} h_X(\tau)||_F^2$ *is the loss where the data distribution $P_X$ is pushed forward by $\mathcal{L}_r \in SO(3)$.*

Prop. 4.7 implies that when $h_X$ (7) is combined with the equivariant components developed in Sec. 4.2, the following three data augmentations are automatically incorporated into the training process: 1) random rotation of each input piece $X_i$, 2) random permutation of the order of the input pieces, and 3) random rotation of the assembled shape.

## 4.4 SAMPLING VIA THE RUNGE-KUTTA METHOD

Finally, when the vector field $v_X$ (4) is learned, we can obtain a sample $\boldsymbol{g}_1$ from $P_X$ by numerically integrating $v_X$ starting from a noise $\boldsymbol{g}_0$ from $P_0$. In this work, we use the Runge-Kutta (RK) solver on $SE(3)^N$, which is a generalization of the classical RK solver on Euclidean spaces. For clarity, we present the formulations below, and refer the readers to Crouch and Grossman (1993) for more details.

To apply the RK method, we first discretize the time interval $[0, 1]$ into $I$ steps, *i.e.*, $\tau_i = \frac{i}{I}$ for $i = 0, \ldots, I$, with a step length $\eta = \frac{1}{I}$. For the given input $X$, denote $f(\boldsymbol{g}X)$ at time $\tau$ by $f_\tau(\boldsymbol{g})$ for simplicity. The first-order RK method (RK1), *i.e.*, the Euler method, is to iterate: $\boldsymbol{g}_{i+1} = \exp(\eta f_{\tau_i}(\boldsymbol{g}_i))\boldsymbol{g}_i$, for $i = 0, \ldots, I$. To achieve higher accuracy, we can use the fourth-order RK method (RK4). More details can be found in E.

## 5 IMPLEMENTATION

This section provides the details of the network $f$ (5). Our design principle is to imitate the standard transformer structure (Vaswani et al., 2017) to retain its best practices. In addition, according to Prop. 4.6, we also require $f$ to be permutation-equivariant and $SO(3)$-equivariant.

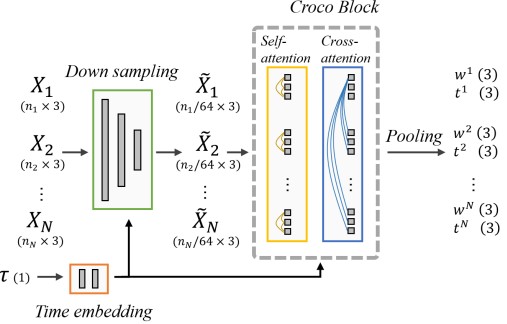

Figure 1: An overview of our model. The shapes of variables are shown in the brackets.

The overall structure of the proposed network is shown in Fig. 1. In a forward pass, the input PC pieces $\{X_i\}_{i=1}^N$ are first downsampled using a few downsampling blocks, and then fed into the Croco blocks (Weinzaepfel et al., 2022) to model their relations. Meanwhile, the time step $\tau$ is first embedded using a multi-layer perceptron (MLP) and then incorporated into the above blocks via adaptive normalization (Peebles and Xie, 2023). The output is finally obtained by a piece-wise pooling.

Next, we provide details of the equivariant attention layers, which are the major components of both the downsampling block and the Croco block, in Sec. 5.1. Other layers, including the nonlinear and normalization layers, are described in Sec. 5.2.

## 5.1 EQUIVARIANT ATTENTION LAYERS

The equivariant attention layers are based on e3nn (Geiger and Smidt, 2022). For the input point cloud, the KNN graph is first built, and the query $Q$, key $K$ and value $V$ matrices are computed for each node. Then the dot-product attention is computed where each node attends to its neighbors. We further use the reduction technique (Passaro and Zitnick, 2023) to accelerate the computation. More details can be found in Appx. G.

Following Croco (Weinzaepfel et al., 2022), we stack two types of attention layers, *i.e.*, the self-attention layer and the cross-attention layer, into a Croco block to learn the features of each PC

piece while incorporating information from other pieces. For self-attention layers, we build KNN graph where the neighbors are selected from the same pieces, and for cross-attention layers, we build KNN graph where the neighbors are selected from the different pieces. In addition, to reduce the computational cost, we use downsampling layers to reduce the number of points before the Croco layers. Each downsampling layer consists of a farthest point sampling (FPS) layer and a self-attention layer.

## 5.2 Adaptive normalization and nonlinear layers

Following the common practice (Devlin et al., 2019), we seek to use the GELU activation function (Hendrycks and Gimpel, 2016) in our transformer structure. However, GELU in its original form is not $SO(3)$-equivariant. To address this issue, we adopt a projection formulation similar to Deng et al. (2021). Specifically, we define the equivariant GELU (Elu) layer as: $Elu(F^l) = GELU(\langle F^l, \widehat{WF^l}\rangle)$ where $\widehat{x} = x/\|x\|$ is the normalization, $W \in \mathbb{R}^{c \times c}$ is a learnable weight. Note that Elu is a natural extension of GELU, because when $l = 0$, $Elu(F^0) = GELU(\pm F^0)$.

As for the normalization layers, we use RMS-type layer normalization layers (Zhang and Sennrich, 2019) following Liao et al. (2023), and we use the adaptive normalization (Peebles and Xie, 2023) technique to incorporate the time step $\tau$. Specifically, we use the adaptive normalization layer *AN* defined as: $AN(F^l, \tau) = F^l/\sigma \cdot MLP(\tau)$, where $\sigma = \sqrt{\frac{1}{c \cdot l_{max}} \sum_{l=1}^{l_{max}} \frac{1}{2l+1} \langle F^l, F^l \rangle}$, $l_{max}$ is the maximum degree, and *MLP* is a multi-layer perceptron that maps $\tau$ to a vector of length $c$.

We finally remark that the network $\boldsymbol{f}$ defined in this section is $SO(3)$-equivariant because each layer is $SO(3)$-equivariant by construction. $\boldsymbol{f}$ is also permutation-equivariant because it does not use any order information of $X_i$.

## 6 Experiment

This section evaluates Eda on practical assembly tasks. After introducing the experiment settings in Sec. 6.1, we first evaluate Eda on the pair-wise registration tasks in Sec. 6.2, and then we consider the multi-piece assembly tasks in Sec. 6.3. An ablation study is finally presented in Sec. 6.4.

## 6.1 Experiment settings

We evaluate the accuracy of an assembly solution using the averaged pair-wise error. For a predicted assembly $\boldsymbol{g}$ and the ground truth $\hat{\boldsymbol{g}}$, the rotation error $\Delta r$ and the translation error $\Delta t$ are computed as: $(\Delta r, \Delta t) = \frac{1}{N(N-1)} \sum_{i \neq j} \tilde{\Delta}(\hat{g}_i, \hat{g}_j g_j^{-1} g_i)$, where the pair-wise error $\tilde{\Delta}$ is computed as $\tilde{\Delta}(g, \hat{g}) = \left(\frac{180}{\pi} accos\left(\frac{1}{2}\left(tr(r\hat{r}^T) - 1\right)\right), \|\hat{t} - t\|\right)$. Here $g = (r, t)$, $\hat{g} = (\hat{r}, \hat{t})$, and $tr(\cdot)$ represents the trace. This metric is the pair-wise rotation/translation error: it measures the averaged error of $\boldsymbol{g}_i$ w.r.t. $\boldsymbol{g}_j$ for all $(i, j)$ pairs of pieces.

For Eda, we use 2 Croco blocks, and 4 downsampling layers with a downsampling ratio 0.25. We use $k = 10$ nearest neighbors, $l_{max} = 2$ degree features with $d = 64$ channels and 4 attention heads. Following Peebles and Xie (2023), we keep an exponential moving average (EMA) with a decay of 0.99, and we use the AdamW (Loshchilov and Hutter, 2017) optimizer with a learning rate $10^{-4}$. Following Esser et al. (2024), we use a logit-normal sampling for time variable $\tau$. For each experiment, we train Eda on 3 Nvidia A100 GPUs for at most 5 days. We denote Eda with $q$ steps of RK$p$ as "Eda (RK$p$, $q$)" , *e.g.*, Eda (RK1, 10) represents Eda with 10 steps of RK1.

## 6.2 Pair-wise registration

This section evaluates Eda on rotated 3DMatch (Zeng et al., 2017) (3DM) dataset containing PC pairs from indoor scenes. Following Huang et al. (2021), we consider the 3DLoMatch split (3DL), which contains PC pairs with smaller overlap ratios. Furthermore, to highlight the ability of Eda on non-overlapped

Table 1: The overlap ratio of PC pairs (%).

|  | 3DM | 3DL | 3DZ |
|---|---|---|---|
| Training set | $(10, 100)$ | | 0 |
| Test set | $(30, 100)$ | $(10, 30)$ | 0 |

pairs with smaller overlap ratios. Furthermore, to highlight the ability of Eda on non-overlapped

assembly tasks, we consider a new split called 3DZeroMatch (3DZ), which contains non-overlapped PC pairs. The comparison of these three splits is shown in Tab. 1.

We compare Eda against the following baseline methods: FGR (Zhou et al., 2016), GEO (Qin et al., 2022), ROI (Yu et al., 2023), and AMR (Chen et al., 2025), where FGR is a classic optimization-based method, GEO and ROI are correspondence-based methods, and AMR is a recently proposed diffusion-like method based on GEO. We report the results of the baseline methods using their official implementations. Note that the correspondence-free methods like Ryu et al. (2024); Wang and Jörnsten (2024) do not scale to this dataset.

Table 2: Quantitative results on rotated 3DMatch. ROI (n): ROI with $n$ RANSAC samples.

|  | 3DM | | 3DL | | 3DZ | |
|---|---|---|---|---|---|---|
|  | $\Delta r$ | $\Delta t$ | $\Delta r$ | $\Delta t$ | $\Delta r$ | $\Delta t$ |
| FGR | 69.5 | 0.6 | 117.3 | 1.3 | – | – |
| GEO | 7.43 | 0.19 | 28.38 | 0.69 | – | – |
| ROI (500) | 5.64 | 0.15 | 21.94 | 0.53 | – | – |
| ROI (5000) | 5.44 | 0.15 | 22.17 | 0.53 | – | – |
| AMR | 5.0 | **0.13** | 20.5 | 0.53 | – | – |
| Eda (RK4, 50) | **2.38** | 0.17 | **8.57** | **0.4** | 78.32 | 2.74 |

We report the results in Tab 2. On 3DM and 3DL, we observe that Eda outperforms the baseline methods by a large margin, especially for rotation errors, where Eda achieves more than $50\%$ lower rotation errors on both 3DL and 3DM. We provide more details of Eda on 3DL in Fig. 5 in the appendix.

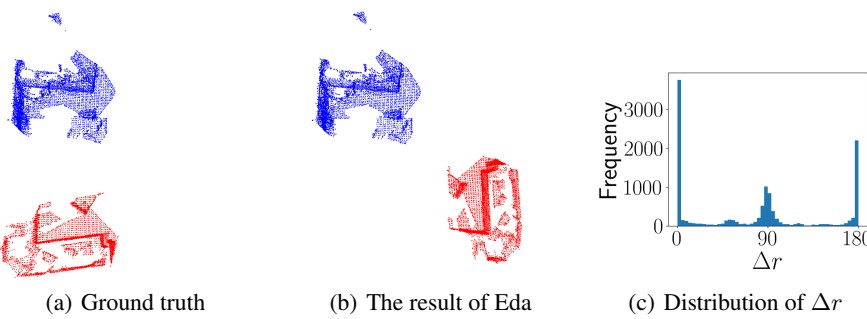

(a) Ground truth          (b) The result of Eda          (c) Distribution of $\Delta r$

Figure 2: More details of Eda on 3DZ. (b): A result of Eda. Cameras are set to look at the room from above. Two PC pieces are marked by different colors. (c): the distribution of $\Delta r$ on the test set.

As for 3DZ, we only report the results of Eda in Tab 2, because all baseline methods are not applicable to 3DZ, *i.e.*, their training goal is undefined when the correspondence does not exist. We observe that Eda's error on 3DZ is much larger compared to that on 3DL, suggesting that there exists much larger ambiguity. Nevertheless, as shown in in Fig. 2(b), Eda indeed learned the global geometry of the indoor scenes instead of just random guessing, because it tends to place large planes, *i.e.*, walls, floors and ceilings, in a parallel or orthogonal position, and keep a plausible distance between walls of the assembled room.

To show that this behavior is consistent in the whole test set, we present the distribution of $\Delta r$ of Eda on 3DZ in Fig. 2(c). A simple intuition is that for rooms consisting of 6 parallel or orthogonal planes (four walls, a floor and a ceiling), if the orthogonality or parallelism of planes is correctly maintained in the assembly, then $\Delta r$ should be 0, 90, or 180. We observe that this is indeed the case in Fig. 2(c), where $\Delta r$ is centered at 0, 90, and 180. We remark that the ability to learn global geometric properties beyond correspondences is a key advantage of Eda, and it partially explains the superior performance of Eda in Tab. 2

### 6.3 MULTI-PIECE ASSEMBLY

This section evaluates Eda on the volume constrained version of BB dataset (Sellán et al., 2022). We consider the shapes with $2 \leq N \leq 8$ pieces in the "everyday" subset. We compare Eda against the following baseline methods: DGL (Zhan et al., 2020), LEV (Wu et al., 2023a), GLO (Sellán et al., 2022), JIG (Lu et al., 2023) and GARF (Li et al., 2025). JIG is correspondence-based, GARF is diffusion-based, and other baseline methods are regression-based. For Eda, we process all fragments by grid downsampling with a grid size 0.02. For the baseline methods, we follow their original preprocessing steps. We do not pretrain GARF for fair comparison,. To reproduce the results of the

baseline methods, we use the implementation of DGL and GLO in the official benchmark suite of BB, and we use the official implementation of LEV, JIG and GARF.

The results are shown in Tab. 3, where we also report the computation time of all methods on the test set on a Nvidia T4 GPU except GARF, which is measured on a A40 GPU because it does not support the T4 GPU. We observe that Eda outperforms all baseline methods by a large margin at a moderate computation cost. We present some qualitative results in Fig. 7 in the appendix, where we observe that Eda can generally reconstruct the shapes more accurately than the baseline methods. An example of the assembly process of Eda is presented in Fig. 3.

Table 3: Quantitative results on BB dataset and the total computation time on the test set.

|  | $\Delta r$ | $\Delta t$ | Time (min) |
|---|---|---|---|
| GLO | 126.3 | 0.3 | **0.9** |
| DGL | 125.8 | 0.3 | **0.9** |
| LEV | 125.9 | 0.3 | 8.1 |
| JIG | 106.5 | 0.24 | 122.2 |
| GARF | 95.6 | 0.2 | (48) |
| Eda (RK1, 10) | 80.64 | **0.16** | 19.4 |
| Eda (RK4, 10) | **79.2** | **0.16** | 76.9 |

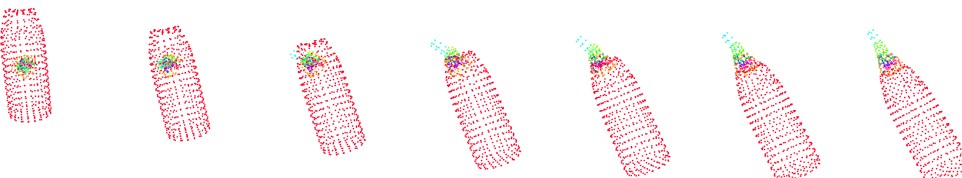

Figure 3: From left to right: the assembly process of a 8-piece bottle by Eda.

### 6.4 ABLATION STUDIES

We first investigate the influence of the number of pieces on the performance of Eda. We use the kitti odometry dataset (Geiger et al., 2012) containing PCs of city road views. For each sequence of data, we keep pieces that are at least 100 meters apart so that they do not necessarily overlap, and we downsample them using grid downsampling with a grid size 0.5. We train Eda on all consecutive pieces of length $2 \sim N_{max}$ in sequences $0 \sim 8$. We call the trained model Eda-$N_{max}$. We then evaluate Eda-$N_{max}$ on all consecutive pieces of length $M$ in sequence $9 \sim 10$.

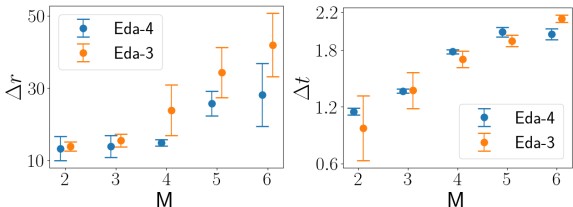

Figure 4: The results of Eda on different number of pieces.

The results are shown in Fig. 4. We observe that for $\Delta r$, when the length of the test data is seen in the training set, i.e., $M \leq N_{max}$, Eda performs well, and $M > N_{max}$ leads to worse performance. In addition, Eda-4 generalizes better than Eda-3 on data of unseen length (5 and 6). The result indicates the necessity of using training data whose lengths subsume that of the test data. Meanwhile, the translation errors of Eda-4 and Eda-3 are comparable, and they both increase with the length of data.

Then we investigate the influence of the components in our theory. We compare Eda with Eda-$O$ on the 3DL dataset, where $O$ is a combination of the following modifications: 1) $r$: removing $r^*$ in $h_X$ (7). 2) $h$: replacing $h_X$ (7) by the canonical path $\overline{h}$. 3) $e$: replacing $f$ by a non-equivariant network. The results are shown in Tab. 4, where we observe that $r$ leads to a small performance drop, while $h$ and $e$ lead to large performance drops. In addition, Eda-$(r, h, e)$ fails to converge. More details can be found in Appx. H.

Table 4: Ablation study.

|  | $\Delta r$ | $\Delta t$ |
|---|---|---|
| Eda | 13.3 | 0.2 |
| Eda-$(r)$ | 15.4 | 0.23 |
| Eda-$(r, h)$ | 79.4 | 0.51 |
| Eda-$(r, e)$ | 86.2 | 0.37 |
| Eda-$(r, h, e)$ | − | − |

## 7 CONCLUSION

This work studied the theory of equivariant flow matching, and presented a multi-piece assembly method, called Eda, based on the theory. We show that Eda can accurately assemble PCs on practical datasets. More discussions can be found in Appx. I.

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

## A   THE USE OF LARGE LANGUAGE MODELS (LLM)

We use an LLM to correct grammar errors.

## B   MORE DETAILS OF THE RELATED TASKS

The registration task aims to reconstruct the scene from multiple overlapped views. A registration method generally consists of two stages: first, each pair of pieces is aligned using a pair-wise method (Qin et al., 2022), then all pieces are merged into a complete shape using a synchronization method (Arrigoni et al., 2016; Lee and Civera, 2022; Gojcic et al., 2020). In contrast to other tasks, the registration task generally assumes that the pieces are overlapped. In other words, it assumes that some points observed in one piece are also observed in the other piece, and the goal is to match the points observed in both pieces, *i.e.*, corresponding points. The state-of-the-art registration methods usually infer the correspondences based on the feature similarity (Yu et al., 2023) learned by neural networks, and then align them using the SVD projection (Arun et al., 1987) or RANSAC.

The robotic manipulation task aims to move one PC to a certain position relative to another PC. For example, one PC can be a cup, and the other PC can be a table, and the goal is to move the cup onto the table. Since the input PCs are sampled from different objects, they are generally non-overlapped. Unlike the other two tasks, this task is generally formulated in a probabilistic setting, as the solution is generally not unique. Various probabilistic models, such as energy-based models (Simeonov et al., 2022; Ryu et al., 2023), or diffusion models (Ryu et al., 2024), have been used for this task.

The reassembly task aims to reconstruct the complete object from multiple fragment pieces. This task is similar to the registration task, except that the input PCs are sampled from different fragments, thus they are not necessarily overlapped, *e.g.*, due to missing pieces or the erosion of the surfaces. Most of the existing methods are based on regression, where the solution is directly predicted from the input PCs (Wu et al., 2023a; Chen et al., 2022; Wang and Jörnsten, 2024). Some probabilistic methods, such as diffusion-based methods (Xu et al., 2024; Scarpellini et al., 2024), have also been proposed. Note that there exist some exceptions (Lu et al., 2023) which assume the overlap of the pieces, and they rely on the inferred correspondences as the registration methods.

A comparison of these three tasks is presented in Tab. 5.

Table 5: Comparison between registration, reassembly and manipulation tasks.

| Task | Number of pieces | Probabilistic/Deterministic | Overlap |
|---|---|---|---|
| Registration | $\geq 2$ | Deterministic | Overlapped |
| Reassembly | $\geq 2$ | Deterministic | Non-overlapped |
| Manipulation | 2 | Probabilistic | Non-overlapped |
| Assembly (this work) | $\geq 2$ | Probabilistic | Non-overlapped |

## C   A WALK-THROUGH OF THE MAIN THEORY

This section provides a walk-through of the theory using the two-piece deterministic example. We follow the notation in example 3.2: let $(r_1, r_2)$ be the solution for the input point clouds $(X_1, X_2)$, meaning $r_1 X_1$ and $r_2 X_2$ are assembled.

Our theory addresses the following equivariance question. Assume that a diffusion model works for the input $(X_1, X_2)$, *i.e.*, the predicted vector field $v_{(X_1,X_2)}$ flows to the correct solution $(r_1, r_2)$. **How to ensure it also works for the perturbed input?** For example, for $SO(3)^2$-equivariance, the question is how to ensure the model also works for $(r_3 X_1, r_4 X_2)$. *i.e.*, to ensure the predicted vector field $v_{(r_3 X_1, r_4 X_2)}$ flows to $(r_1 r_3^{-1}, r_2 r_4^{-1})$.

**Corollary 4.4** shows that the goal can be achieved if $v_{(r_3 X, r_4 X_2)}$ is a proper "transformation" of $v_{(X_1,X_2)}$ (relatedness), and the noise is invariant.

Then, the next question is how to satisfy the relatedness requirement. **Proposition 4.5** suggests that this can be simply done by parametrizing the vector fields as

$$v_{(X_1,X_2)}(r_7, r_8) = f(r_7 X_1, r_8 X_2)(r_7 \oplus r_8), \quad where \quad f(X_1, X_2) = (w_1, t_1) \oplus (w_2, t_2) \quad (8)$$

is a neural network mapping $(X_1, X_2)$ to their respective rotation/translation velocity components $w$ and $t$, and $\oplus$ is the concatenation. In summary, we can now answer the question from the last paragraph: if the diffusion model predicts the vector field as in (8) and it works for $(X_1, X_2)$, then it also works for $(r_3 X_1, r_4 X_2)$.

Further more, **Proposition 4.6** suggests that, to ensure the other two requirements (permutation equivariance and SO(3)-invariance) of the model, $f$ needs to satisfy

$$f(X_2, X_1) = (w_2, t_2) \oplus (w_1, t_1) \quad and \quad f(rX_1, rX_2) = (rw_1, rt_1) \oplus (rw_2, rt_2) \quad (9)$$

Finally, **Proposition 4.7** suggests that some data augmentations are not needed when all the above requirements are satisfied. For example, for data $(X_1, X_2)$ we learn a vector field $v_{(X_1,X_2)}$. We can use randomly augmented data $(r_3 X_1, r_4 X_2)$ and learn $v_{(r_3 X_1, r_4 X_2)}$. However, this is not necessary because $v_{(r_3 X_1, r_4 X_2)}$ is already guaranteed to be a transformation of $v_{(X_1,X_2)}$ as described above, and the loss for them is the same, *i.e.*, learning $v_{(X_1,X_2)}$ alone is enough. Similar results hold for the other two types of augmentations.

# D  CONNECTIONS WITH BI-EQUIVARIANCE

This section briefly discusses the connections between Def. 3.1 and the equivariances defined in Ryu et al. (2024) and Wang and Jörnsten (2024) in pair-wise assembly tasks.

We first recall the definition of the probabilistic bi-equivariance.

**Definition D.1** (Eqn. (10) in Ryu et al. (2024) and Def. (1) in Ryu et al. (2022)). $\hat{P} \in \mu(SE(3))$ is bi-equivariant if for all $g_1, g_2 \in SO(3)$, PCs $X_1, X_2$, and a measurable set $A \subseteq SE(3)$,

$$\hat{P}(A|X_1, X_2) = \hat{P}(g_2 A g_1^{-1}|g_1 X_1, g_2 X_2). \quad (10)$$

Note that we only consider $g_1, g_2 \in SO(3)$ instead of $g_1, g_2 \in SE(3)$ because we require all input PCs, *i.e.*, $X_i, g_i X_i, i = 1, 2$, to be centered.

Then we recall Def. 3.1 for pair-wise assembly tasks:

**Definition D.2** (Restate $SO(3)^2$-equivariance and $SO(3)$-invariance in Def. 3.1 for pair-wise problems). Let $X_1, X_2$ be the input PCs and $P \in \mu(SE(3) \times SE(3))$.

- $P$ is $SO(3)^2$-equivariant if $P(A|X_1, X_2) = P(A(g_1^{-1}, g_2^{-1})|g_1 X_1, g_2 X_2)$ for all $g_1, g_2 \in SO(3)$ and $A \subseteq SO(3) \times SO(3)$, where $A(g_1^{-1}, g_2^{-1}) = \{(a_1 g_1^{-1}, a_2 g_2^{-1}) : (a_1, a_2) \in A\}$.

- $P$ is $SO(3)$-invariant if $P(A|X_1, X_2) = P(rA|X_1, X_2)$ for all $r \in SO(3)$ and $A \subseteq SO(3) \times SO(3)$.

Intuitively, both Def. D.1 and Def. D.2 describe the equivariance property of an assembly solution, and the only difference is that Def. D.1 describes the special case where $X_1$ can be rigidly transformed and $X_2$ is fixed, while Def. D.2 describes the solution where both $X_1$ and $X_2$ can be rigidly transformed. In other words, a solution satisfying Def. D.2 can be converted to a solution satisfying Def. D.1 by fixing $X_2$. Formally, we have the following proposition.

**Proposition D.3.** *Let $P$ be $SO(3)^2$-equivariant and $SO(3)$-invariant. If $\tilde{P}(A|X_1, X_2) \triangleq P(A \times \{e\}|X_1, X_2)$ for $A \subseteq SO(3)$, then $\tilde{P}$ is bi-equivariant.*

*Proof.* We prove this proposition by directly verifying the definition.

$$\tilde{P}(g_2 A g_1^{-1}|g_1 X_1, g_2 X_2) = P(g_2 A g_1^{-1} \times \{e\}|g_1 X_1, g_2 X_2) \quad (11)$$

$$= P(g_2 A \times \{e\}|X_1, g_2 X_2) \quad (12)$$

$$= P(A \times \{g_2^{-1}\}|X_1, g_2 X_2) \quad (13)$$

$$= P(A \times \{e\}|X_1, X_2) \quad (14)$$

$$= \tilde{P}(A|X_1, X_2). \quad (15)$$

Here, the second and the fourth equation hold because $P$ is $SO(3)^2$-equivariant, the third equation holds because $P$ is $SO(3)$-invariant, and the first and last equation are due to the definition. $\qquad\square$

We note that the deterministic definition of bi-equivariance in Wang and Jörnsten (2024) is a special case of Def. D.1, where $\hat{P}$ is a Dirac delta function. In addition, as discussed in Appx. E in Wang and Jörnsten (2024), a major limitation of the deterministic definition of bi-equivariance is that it cannot handle symmetric shapes. In contrast, it is straightforward to see that the probabilistic definition, *i.e.*, both Def. D.1 and Def. D.2 are free from this issue. Here, we consider the example in Wang and Jörnsten (2024). Assume that $X_1$ is symmetric, *i.e.*, there exists $g_1 \in SO(3)$ such that $g_1 X_1 = X_1$. Under Def. D.1, we have $P(A|X_1, X_2) = P(A|g_1 X_1, X_2) = P(Ag_1|X_1, X_2)$, which simply means that $P(A|X_1, X_2)$ is $\mathcal{R}_{g_1}$-invariant. Note that this will not cause any contradiction, *i.e.*, the feasible set is not empty. For example, a uniform distribution on $SO(3)$ is $\mathcal{R}_{g_1}$-invariant.

As for the permutation-equivariance, the swap-equivariance in Wang and Jörnsten (2024) is a deterministic pair-wise version of the permutation-equivariance in Def. D.2, and they both mean that the assembled shape is independent of the order of the input pieces.

# E  THE RK4 FORMULATION

$$k_1 = f_{\tau_i}(\boldsymbol{g}_i),\ k_2 = f_{\tau_i + \frac{1}{2}\eta}\big(\exp(\frac{1}{2}\eta k_1)\boldsymbol{g}_i\big),\ k_3 = f_{\tau_i + \frac{1}{2}\eta}\big(\exp(\frac{1}{2}\eta k_2)\boldsymbol{g}_i\big),\ k_4 = f_{\tau_i + \eta}\big(\exp(\eta k_3)\boldsymbol{g}_i\big),$$

$$\boldsymbol{g}_{i+1} = \exp(\frac{1}{6}\eta k_4)\exp(\frac{1}{3}\eta k_3)\exp(\frac{1}{3}\eta k_2)\exp(\frac{1}{6}\eta k_1)\boldsymbol{g}_i. \tag{16}$$

Note that RK4 (16) is more computationally expensive than RK1, because it requires four evaluations of $v_X$ at different points at each step, *i.e.*, four forward passes of network $f$, while the Euler method only requires one evaluation per step.

# F  PROOFS

## F.1  PROOF IN SEC. 4.2

To prove Thm. 4.2, which established the relations between related vector fields and equivariant distributions, we proceed in two steps: first, we prove lemma F.1, which connects related vector fields to equivariant mappings; then we prove lemma. F.2, which connects equivariant mappings to equivariant distributions.

**Lemma F.1.** *Let $G$ be a smooth manifold, $F : G \to G$ be a diffeomorphism. If vector field $v_\tau$ is $F$-related to vector field $w_\tau$ for $\tau \in [0, 1]$, then $F \circ \phi_\tau = \psi_\tau \circ F$, where $\phi_\tau$ and $\psi_\tau$ are generated by $v_\tau$ and $w_\tau$ respectively.*

*Proof.* Let $\tilde{\psi}_\tau \triangleq F \circ \phi_\tau \circ F^{-1}$. We only need to show that $\tilde{\psi}_\tau$ coincides with $\psi_\tau$.

We consider a curve $\tilde{\psi}_\tau(F(\boldsymbol{g}_0))$, $\tau \in [0, 1]$, for a arbitrary $\boldsymbol{g}_0 \in G$. We first verify that $\tilde{\psi}_0(F(\boldsymbol{g}_0)) = F \circ \phi_0 \circ F^{-1} \circ F(\boldsymbol{g}_0) = F(\boldsymbol{g}_0)$. Note that the second equation holds because $\phi_0(\boldsymbol{g}_0) = \boldsymbol{g}_0$, *i.e.*, $\phi_\tau$ is an integral path. Then we verify

$$\frac{\partial}{\partial \tau}(\tilde{\psi}_\tau(F(\boldsymbol{g}_0))) = \frac{\partial}{\partial \tau}(F \circ \phi_\tau(\boldsymbol{g}_0)) \tag{17}$$

$$= F_{*, \phi_\tau(\boldsymbol{g}_0)} \circ \frac{\partial}{\partial \tau}(\phi_\tau(\boldsymbol{g}_0)) \tag{18}$$

$$= F_{*, \phi_\tau(\boldsymbol{g}_0)} \circ v_\tau(\phi_\tau(\boldsymbol{g}_0)) \tag{19}$$

$$= w_\tau(F \circ \phi_\tau(\boldsymbol{g}_0)) \tag{20}$$

$$= w_\tau(\tilde{\psi}_\tau(F(\boldsymbol{g}_0))) \tag{21}$$

where the 2-nd equation holds due to the chain rule, and the 4-th equation holds becomes $v_\tau$ is $F$-related to $w_\tau$. Therefore, we can conclude that $\tilde{\psi}_\tau(F(\boldsymbol{g}_0))$ is an integral curve generated by $w_\tau$

starting from $F(\boldsymbol{g}_0)$. However, by definition of $\psi_\tau$, $\psi_\tau(F(\boldsymbol{g}_0))$ is also the integral curve generated by $w_\tau$ and starts from $F(\boldsymbol{g}_0)$. Due to the uniqueness of integral curves, we have $\tilde{\psi}_\tau = \psi_\tau$. $\qquad\square$

**Lemma F.2.** *Let $\phi$, $\psi$, $F : G \to G$ be three diffeomorphisms satisfying $F \circ \phi = \psi \circ F$. We have $F_\#(\phi_\#\rho) = \psi_\#(F_\#\rho)$ for all distribution $\rho$ on $G$.*

*Proof.* Let $A \subseteq G$ be a measurable set. We first verify that $\phi^{-1}(F^{-1}(A)) = F^{-1}(\psi^{-1}(A))$: If $x \in \phi^{-1}(F^{-1}(A))$, then $(F \circ \phi)(x) \in A$. Since $F \circ \phi = \psi \circ F$, we have $(\psi \circ F)(x) \in A$, which implies $x \in F^{-1}(\psi^{-1}(A))$, *i.e.*, $\phi^{-1}(F^{-1}(A)) \subseteq F^{-1}(\psi^{-1}(A))$. The other side can be verified similarly. Then we have

$$(F_\#(\phi_\#\rho))(A) = \rho(\phi^{-1}(F^{-1}(A))) = \rho(F^{-1}(\psi^{-1}(A))) = (\psi_\#(F_\#\rho))(A), \qquad (22)$$

which proves the lemma. $\qquad\square$

Now, we can prove Thm. 4.2 using the above two lemmas.

*Proof of Thm. 4.2.* Since $v_X$ is $F$-related to $v_Y$, according to lemma F.1, we have $F \circ \phi_X = \phi_Y \circ F$. Then according to lemma F.2, we have $F_\#(\phi_{X\#}P_0) = \phi_{Y\#}(F_\#P_0)$. The proof is complete by letting $P_X = \phi_{X\#}P_0$ and $P_Y = \phi_{Y\#}(F_\#P_0)$. $\qquad\square$

We remark that our theory extends the results in Köhler et al. (2020), where only invariance is considered, Specifically, we have the following corollary.

**Corollary F.3** (Thm 2 in Köhler et al. (2020)). *Let $G$ be the Euclidean space, $F$ be a diffeomorphism on $G$, and $v_\tau$ be a $F$-invariant vector field, i.e., $v_\tau$ is $F$-related to $v_\tau$, then we have $F \circ \phi_\tau = \phi_\tau \circ F$, where $\phi_\tau$ is generated by $v_\tau$.*

*Proof.* This is a direct consequence of lemma. F.1 where $G$ is the Euclidean space and $w_\tau = v_\tau$. $\quad\square$

Note that the terminology used in Köhler et al. (2020) is different from ours: The $F$-invariant vector fields in our work is called $F$-equivariant vector field in Köhler et al. (2020), and Köhler et al. (2020) does not consider general related vector fields.

Finally, we present the proof of Prop. 4.5 and Prop. 4.6.

*Proof of Prop. 4.5.* If $v_X$ is $\mathcal{R}_{\boldsymbol{g}^{-1}}$-related to $v_{\boldsymbol{g}X}$, we have $v_{\boldsymbol{g}X}(\hat{\boldsymbol{g}}\boldsymbol{g}^{-1}) = (\mathcal{R}_{\boldsymbol{g}^{-1}})_{*,\hat{\boldsymbol{g}}}v_X(\hat{\boldsymbol{g}})$ for all $\hat{\boldsymbol{g}}, \boldsymbol{g} \in SE(3)^N$. By letting $\boldsymbol{g} = \hat{\boldsymbol{g}}$, we have

$$v_X(\boldsymbol{g}) = (\mathcal{R}_{\boldsymbol{g}})_{*,e}v_{\boldsymbol{g}X}(e) \qquad (23)$$

where $(\mathcal{R}_{\boldsymbol{g}})_{*,e} = \left((\mathcal{R}_{\boldsymbol{g}^{-1}})_{*,\boldsymbol{g}}\right)^{-1}$ due to the chain rule of $\mathcal{R}_{\boldsymbol{g}}\mathcal{R}_{\boldsymbol{g}^{-1}} = e$.

On the other hand, if Eqn. (23) holds, we have

$$(\mathcal{R}_{\boldsymbol{g}^{-1}})_{*,\hat{\boldsymbol{g}}}v_X(\hat{\boldsymbol{g}}) = (\mathcal{R}_{\boldsymbol{g}^{-1}})_{*,\hat{\boldsymbol{g}}}(\mathcal{R}_{\hat{\boldsymbol{g}}})_{*,e}v_{\hat{\boldsymbol{g}}X}(e) = (\mathcal{R}_{\hat{\boldsymbol{g}}\boldsymbol{g}^{-1}})_{*,e}v_{\hat{\boldsymbol{g}}X}(e) = v_{\boldsymbol{g}X}(\hat{\boldsymbol{g}}\boldsymbol{g}^{-1}), \qquad (24)$$

which suggests that $v_X$ is $\mathcal{R}_{\boldsymbol{g}^{-1}}$-related to $v_{\boldsymbol{g}X}$. Note that the second equation holds due to the chain rule of $\mathcal{R}_{\boldsymbol{g}^{-1}}\mathcal{R}_{\hat{\boldsymbol{g}}} = \mathcal{R}_{\hat{\boldsymbol{g}}\boldsymbol{g}^{-1}}$, and the first and the third equation are the result of Eqn. (23). $\qquad\square$

*Proof of Prop. 4.6.* 1) Assume $v_X$ is $\sigma$-related to $v_{\sigma X}$: $(\sigma)_{*,\boldsymbol{g}}v_X(\boldsymbol{g}) = V_{\sigma X}(\sigma(\boldsymbol{g}))$. By inserting Eqn. (5) to this equation, we have

$$(\sigma)_{*,\boldsymbol{g}}(\mathcal{R}_{\boldsymbol{g}})_{*,e}f(\boldsymbol{g}X) = (\mathcal{R}_{\sigma\boldsymbol{g}})_{*,e}f(\sigma(\boldsymbol{g})\sigma(X)). \qquad (25)$$

Since $\sigma \circ \mathcal{R}_{\boldsymbol{g}} = \mathcal{R}_{\sigma\boldsymbol{g}} \circ \sigma$, by the chain rule, we have $\sigma_*(\mathcal{R}_{\boldsymbol{g}})_* = (\mathcal{R}_{\sigma\boldsymbol{g}})_*\sigma_*$. In addition, $\sigma(\boldsymbol{g})\sigma(X) = \sigma(\boldsymbol{g}X)$. Thus, this equation can be simplified as

$$(\mathcal{R}_{\sigma\boldsymbol{g}})_*\sigma_*f(\boldsymbol{g}X) = (\mathcal{R}_{\sigma\boldsymbol{g}})_{*,e}f(\sigma(\boldsymbol{g}X)) \qquad (26)$$

which suggests

$$\sigma_*f = f \circ \sigma. \qquad (27)$$

The first statement in Prop. 4.6 can be proved by reversing the discussion.

2) Assume $v_X$ is $\mathcal{L}_r$-related to $v_X$: $(\mathcal{L}_r)_{*,g}v_X(\boldsymbol{g}) = V_X(r\boldsymbol{g})$. By inserting Eqn. (5) to this equation, we have

$$(\mathcal{L}_r)_{*,g}(\mathcal{R}_{\boldsymbol{g}})_{*,e}f(\boldsymbol{g}X) = (\mathcal{R}_{r\boldsymbol{g}})_{*,e}f(r\boldsymbol{g}X). \tag{28}$$

Since $\mathcal{R}_{r\boldsymbol{g}} = \mathcal{R}_{\boldsymbol{g}} \circ \mathcal{R}_r$, by the chain rule, we have $(\mathcal{R}_{r\boldsymbol{g}})_{*,e} = (\mathcal{R}_{\boldsymbol{g}})_{*,r}(\mathcal{R}_r)_{*,e}$. In addition, $(\mathcal{L}_r)(\mathcal{R}_{\boldsymbol{g}}) = (\mathcal{R}_{\boldsymbol{g}})(\mathcal{L}_r)$, by the chain rule, we have $(\mathcal{L}_r)_{*,\boldsymbol{g}}(\mathcal{R}_{\boldsymbol{g}})_{*,e} = (\mathcal{R}_{\boldsymbol{g}})_{*,r}(\mathcal{L}_r)_{*,e}$. Thus the above equation can be simplified as

$$(\mathcal{L}_r)_{*,e}f(\boldsymbol{g}X) = (\mathcal{R}_r)_{*,e}f(r\boldsymbol{g}X) \tag{29}$$

which implies

$$f \circ r = (\mathcal{R}_{r^{-1}})_{*,r} \circ (\mathcal{L}_r)_{*,e} \circ f. \tag{30}$$

By representing $f$ in the matrix form, we have

$$w_\times^i(rX) = rw_\times^i(X)r^T \tag{31}$$

$$t^i(rX) = rt^i(X) \tag{32}$$

for all $i$, where $r$ on the right hand side represents the matrix form of the rotation $r$. Here the first equation can be equivalently written as $w^i(rX) = rw^i(X)$. The second statement in Prop. 4.6 can be proved by reversing the discussion. □

## F.2 PROOFS IN SEC. 4.3

To establish the results in this section, we need to assume the uniqueness of $r^*$ (6):

**Assumption F.4.** The solution to (6) is unique.

Note that this assumption is mild. A sufficient condition (Wang and Jörnsten, 2024) of assumption F.4 is that the singular values of $\tilde{\boldsymbol{g}}_1^T \boldsymbol{g}_0 \in \mathbb{R}^{3\times3}$ satisfy $\sigma_1 \geq \sigma_2 > \sigma_3 \geq 0$, *i.e.*, $\sigma_2$ and $\sigma_3$ are not equal. We leave the more general treatment without requiring the uniqueness of $r^*$ to future work.

We first justify the definition of $\boldsymbol{g}_1 = r^*\tilde{\boldsymbol{g}}_1$ by showing that $\boldsymbol{g}_1$ follows $P_1$ in the following proposition.

**Proposition F.5.** *Let $P_0$ and $P_1$ be two $SO(3)$-invariant distributions, and $\boldsymbol{g}_0$, $\tilde{\boldsymbol{g}}_1$ be independent samples from $P_0$ and $P_1$ respectively. If $r^*$ is given by (6) and assumption F.4 holds, then $\boldsymbol{g}_1 = r^*\tilde{\boldsymbol{g}}_1$ follows $P_1$.*

*Proof.* Define $A_{\tilde{\boldsymbol{g}}_1} = \{\boldsymbol{g}_0|r^*(\boldsymbol{g}_0, \tilde{\boldsymbol{g}}_1) = e\}$, where we write $r^*$ as a function of $\tilde{\boldsymbol{g}}_1$ and $\boldsymbol{g}_0$. Then we have $P(r^* = e|\tilde{\boldsymbol{g}}_1) = P_0(A_{\tilde{\boldsymbol{g}}_1})$ by definition. In addition, due to the uniqueness of the solution to (6), for an arbitrary $\hat{r} \in SO(3)$, we have $P(r^* = \hat{r}|\tilde{\boldsymbol{g}}_1) = P_0(\hat{r}A_{\tilde{\boldsymbol{g}}_1})$. Since $P_0$ is $SO(3)$-invariant, we have $P_0(\hat{r}A_{\tilde{\boldsymbol{g}}_1}) = P_0(A_{\tilde{\boldsymbol{g}}_1})$, thus, $P(r^* = \hat{r}|\tilde{\boldsymbol{g}}_1) = P(r^* = e|\tilde{\boldsymbol{g}}_1)$. In other words, for a given $\tilde{\boldsymbol{g}}_1$, $r^*$ follows the uniform distribution $U_{SO(3)}$.

Finally we compute the probability density of $\boldsymbol{g}_1$:

$$P(\boldsymbol{g}_1) = \int P(r^* = \hat{r}^{-1}|\hat{r}\boldsymbol{g}_1)P_1(\hat{r}\boldsymbol{g}_1)d\hat{r} \tag{33}$$

$$= \int U_{SO(3)}(\hat{r})P_1(\boldsymbol{g}_1)d\hat{r} \tag{34}$$

$$= P_1(\boldsymbol{g}_1), \tag{35}$$

which suggests that $\boldsymbol{g}_1$ follows $P_1$. Here the second equation holds because $P_1$ is $SO(3)$-invariant. □

Then we discuss the equivariance of the constructed $h_X$ (7).

**Proposition F.6.** *Given $\boldsymbol{r} \in SO(3)^N$, $\boldsymbol{g}_0, \tilde{\boldsymbol{g}}_1 \in SE(3)^N$, $\sigma \in S_N$, $r \in SO(3)$ and $\tau \in [0,1]$. Let $h_X$ be a path generated by $\boldsymbol{g}_0$ and $\tilde{\boldsymbol{g}}_1$. Under assumption F.4,*

- *if $h_{\boldsymbol{r}X}$ is generated by $\boldsymbol{g}_0\boldsymbol{r}^{-1}$ and $\tilde{\boldsymbol{g}}_1\boldsymbol{r}^{-1}$, then $h_{\boldsymbol{r}X}(\tau) = \mathcal{R}_{\boldsymbol{r}^{-1}}h_X(\tau)$.*

- *if $h_{\sigma X}$ is generated by $\sigma(\boldsymbol{g}_0)$ and $\sigma(\tilde{\boldsymbol{g}}_1)$, then $h_{\sigma X}(\tau) = \sigma(h_X(\tau))$.*

- *if $\hat{h}_X$ is generated by $r\boldsymbol{g}_0$ and $r\tilde{\boldsymbol{g}}_1$, then $\hat{h}_X(\tau) = \mathcal{L}_r(h_X(\tau))$.*

*Proof.* 1) Due to the uniqueness of the solution to (6), we have $r^*(\boldsymbol{g}_0\boldsymbol{r}^{-1}, \tilde{\boldsymbol{g}}_1\boldsymbol{r}^{-1}) = r^*(\boldsymbol{g}_0, \tilde{\boldsymbol{g}}_1)$. Thus, we have

$$h_{\boldsymbol{r}X}(\tau) = \exp(\tau \log(\boldsymbol{g}_1\boldsymbol{g}_0^{-1}))\boldsymbol{g}_0\boldsymbol{r}^{-1} = \mathcal{R}_{\boldsymbol{r}^{-1}}(h_{\boldsymbol{r}X}(\tau)). \tag{36}$$

2) Due to the uniqueness of the solution to (6), we have $r^*(\sigma(\boldsymbol{g}_0), \sigma(\tilde{\boldsymbol{g}}_1)) = \sigma(r^*(\boldsymbol{g}_0, \tilde{\boldsymbol{g}}_1))$. Thus, we have $\sigma(h_X) = h_{\sigma X}$.

3) Due to the uniqueness of the solution to (6), we have $r^*(r\boldsymbol{g}_0, r\tilde{\boldsymbol{g}}_1) = rr^*(\boldsymbol{g}_0, \tilde{\boldsymbol{g}}_1)r^{-1}$. Thus,

$$\hat{h}_{\boldsymbol{r}X}(\tau) = \exp(\tau \log(rr^*\tilde{\boldsymbol{g}}_1\boldsymbol{g}_0^{-1}r^{-1}))r\boldsymbol{g}_0 = r\exp(\tau \log(r^*\tilde{\boldsymbol{g}}_1\boldsymbol{g}_0^{-1}))\boldsymbol{g}_0 = \mathcal{L}_r(h_X(\tau)). \tag{37}$$

$\square$

With the above preparation, we can finally prove Prop. 4.7.

*Proof of Prop. 4.7.* 1) By definition

$$L(\boldsymbol{r}X) = \mathbb{E}_{\tau, \boldsymbol{g}_0' \sim P_0, \tilde{\boldsymbol{g}}_1' \sim P_{\boldsymbol{r}X}} ||v_{\boldsymbol{r}X}(h_{\boldsymbol{r}X}(\tau)) - \frac{\partial}{\partial \tau}h_{\boldsymbol{r}X}(\tau)||_F^2, \tag{38}$$

where $h_{\boldsymbol{r}X}$ is the path generated by $\boldsymbol{g}_0'$ and $\tilde{\boldsymbol{g}}_1'$. Since $P_0 = (\mathcal{R}_{\boldsymbol{r}^{-1}})_\# P_0$ and $P_{\boldsymbol{r}X} = (\mathcal{R}_{\boldsymbol{r}^{-1}})_\# P_X$ by assumption, we can write $\boldsymbol{g}_0' = \boldsymbol{g}_0\boldsymbol{r}^{-1}$ and $\tilde{\boldsymbol{g}}_1' = \tilde{\boldsymbol{g}}_1\boldsymbol{r}^{-1}$, where $\boldsymbol{g}_0 \sim P_0$ and $\tilde{\boldsymbol{g}}_1 \sim P_X$. According to the first part of Prop. F.6, we have $h_{\boldsymbol{r}X}(\tau) = \mathcal{R}_{\boldsymbol{r}^{-1}}h_X(\tau)$, where $h_X$ is a path generated by $\boldsymbol{g}_0$ and $\tilde{\boldsymbol{g}}_1$. By taking derivative on both sides of the equation, we have $\frac{\partial}{\partial \tau}h_{\boldsymbol{r}X}(\tau) = (\mathcal{R}_{\boldsymbol{r}^{-1}})_{*, h_X(\tau)}\frac{\partial}{\partial \tau}h_X(\tau)$. Then we have

$$L(\boldsymbol{r}X) = \mathbb{E}_{\tau, \boldsymbol{g}_0' \sim P_0, \tilde{\boldsymbol{g}}_1' \sim P_{\boldsymbol{r}X}} ||v_{\boldsymbol{r}X}(\mathcal{R}_{\boldsymbol{r}^{-1}}h_X(\tau)) - (\mathcal{R}_{\boldsymbol{r}^{-1}})_{*, h_X(\tau)}\frac{\partial}{\partial \tau}h_X(\tau)||_F^2 \tag{39}$$

by inserting these two equations into Eqn. (38). Since $v_X$ is $\mathcal{R}_{\boldsymbol{r}^{-1}}$-related to $v_{\boldsymbol{r}X}$ by assumption, we have $v_{\boldsymbol{r}X}(\mathcal{R}_{\boldsymbol{r}^{-1}}h_X(\tau)) = (\mathcal{R}_{\boldsymbol{r}^{-1}})_{*, h_X(\tau)}v_X(h_X(\tau))$. Thus, we have

$$||v_{\boldsymbol{r}X}(\mathcal{R}_{\boldsymbol{r}^{-1}}h_X(\tau)) - (\mathcal{R}_{\boldsymbol{r}^{-1}})_{*, h_X(\tau)}\frac{\partial}{\partial \tau}h_X(\tau)||_F^2 = ||(\mathcal{R}_{\boldsymbol{r}^{-1}})_{*, h_X(\tau)}(v_{\boldsymbol{r}X}(h_X(\tau)) - \frac{\partial}{\partial \tau}h_X(\tau))||_F^2$$

$$= ||(v_{\boldsymbol{r}X}(h_X(\tau)) - \frac{\partial}{\partial \tau}h_X(\tau))||_F^2 \tag{40}$$

where the second equation holds because $(\mathcal{R}_{\boldsymbol{r}^{-1}})_{*, h_X(\tau)}$ is an orthogonal matrix. The desired result follows.

2) The second statement can be proved similarly as the first one, where $\sigma$-equivariance is considered instead of $\mathcal{R}_{\boldsymbol{r}^{-1}}$-equivariance.

3) Denote $\boldsymbol{g}_0' = r\boldsymbol{g}_0$ and $\tilde{\boldsymbol{g}}_1' = r\tilde{\boldsymbol{g}}_1$, where $\boldsymbol{g}_0 \sim P_0$ and $\tilde{\boldsymbol{g}}_1 \sim P_X$. According to the third part of Prop. F.6, we have $\hat{h}_X(\tau) = \mathcal{L}_r(h_X(\tau))$. By taking derivative on both sides of the equation, we have $\frac{\partial}{\partial \tau}\hat{h}_X(\tau) = (\mathcal{L}_r)_{*, h_X(\tau)}\frac{\partial}{\partial \tau}h_X(\tau)$. Then the rest of the proof can be conducted similarly to the first part of the proof. $\square$

## G MODEL DETAILS

Let $F_u^l \in \mathbb{R}^{c \times (2l+1)}$ be a channel-$c$ degree-$l$ feature at point $u$. The equivariant dot-product attention is defined as:

$$A_u^l = \sum_{v \in KNN(u) \setminus \{u\}} \frac{\exp(\langle Q_u, K_{vu}\rangle)}{\sum_{v' \in KNN(u) \setminus \{u\}} \exp(\langle Q_u, K_{v'u}\rangle)}V_{vu}^l, \tag{41}$$

where $\langle \cdot, \cdot \rangle$ is the dot product, $KNN(u) \subseteq \bigcup_i X_i$ is a subset of points $u$ attends to, $K, V \in \mathbb{R}^{c \times (2l+1)}$ take the form of the e3nn (Geiger and Smidt, 2022) message passing, and $Q \in \mathbb{R}^{c \times (2l+1)}$ is obtained by a linear transform:

$$Q_u = \bigoplus_l W_Q^l F_u^l, \quad K_v = \bigoplus_l \sum_{l_e, l_f} c_K^{(l, l_e, l_f)}(|uv|)Y^{l_e}(\widehat{vu}) \otimes_{l_e, l_f}^l F_v^{l_f}, \tag{42}$$

$$V_v^l = \sum_{l_e, l_f} c_V^{(l, l_e, l_f)}(|uv|)Y^{l_e}(\widehat{vu}) \otimes_{l_e, l_f}^l F_v^{l_f}. \tag{43}$$

Here, $W_Q^l \in \mathbb{R}^{c \times c}$ is a learnable weight, $|vu|$ is the distance between point $v$ and $u$, $\widehat{vu} = \vec{vu}/|vu| \in \mathbb{R}^3$ is the normalized direction, $Y^l : \mathbb{R}^3 \to \mathbb{R}^{2l+1}$ is the degree-$l$ spherical harmonic function, $c : \mathbb{R}_+ \to \mathbb{R}$ is a learnable function that maps $|vu|$ to a coefficient, and $\otimes$ is the tensor product with the Clebsch-Gordan coefficients.

To accelerate the computation of $K$ and $V$, we use the $SO(2)$-reduction technique (Passaro and Zitnick, 2023), which rotates the edge $uv$ to the $y$-axis, so that the computation of spherical harmonic function, the Clebsch-Gordan coefficients, and the iterations of $l_e$ are no longer needed.

The main idea of $SO(2)$-reduction (Passaro and Zitnick, 2023) is to rotate the edge $uv$ to the $y$-axis, and then update node feature in the rotated space. Since all 3D rotations are reduced to 2D rotations about the $y$-axis in the rotated space, the feature update rule is greatly simplified.

Here, we describe this technique in the matrix form to facilitates better parallelization. Let $F_v^l \in \mathbb{R}^{c \times (2l+1)}$ be a $c$-channel $l$-degree feature of point $v$, and $L > 0$ be the maximum degree of features. We construct $\hat{F}_v^l \in \mathbb{R}^{c \times (2L+1)}$ by padding $F_v^l$ with $L - l$ zeros at the beginning and the end of the feature, then we define the full feature $F_v \in \mathbb{R}^{c \times L \times (2L+1)}$ as the concatenate of all $\hat{F}_v^l$ with $0 < l \le L$. For an edge $vu$, there exists a rotation $r_{vu}$ that aligns $uv$ to the $y$-axis. We define $R_{vu} \in \mathbb{R}^{L \times (2L+1) \times (2L+1)}$ to be the full rotation matrix, where the $l$-th slice $R_{vu}[l,:,:]$ is the $l$-th Wigner-D matrix of $r_{vu}$ with zeros padded at the boundary. $K_v$ defined in (42) can be efficiently computed as

$$K_v = R_{vu}^T \times_{1,2} (W_K \times_3 (D_K \times_{1,2} R_{vu} \times_{1,2} F_v)), \tag{44}$$

where $M_1 \times_i M_2$ represents the batch-wise multiplication of $M_1$ and $M_2$ with the $i$-th dimension of $M_2$ treated as the batch dimension. $W_K \in \mathbb{R}^{(cL) \times (cL)}$ is a learnable weight, $D_K \in \mathbb{R}^{c \times (2L+1) \times (2L+1)}$ is a learnable matrix taking the form of 2D rotations about the $y$-axis, *i.e.*, for each $i$, $D_K[i,:,:]$ is

$$\begin{bmatrix} a_1 & & & & & & & & -b_1 \\ & a_2 & & & & & & -b_2 & \\ & & \ddots & & & & \iddots & & \\ & & & a_{L-1} & -b_{L-1} & & & & \\ & & & & a_L & & & & \\ & & & b_{L-1} & a_{L-1} & & & & \\ & & \iddots & & & & \ddots & & \\ & b_2 & & & & & & a_2 & \\ b_1 & & & & & & & & a_1 \end{bmatrix}, \tag{45}$$

where $a_1, \cdots, a_L, b_1, \cdots, b_{L-1} : \mathbb{R}_+ \to \mathbb{R}$ are learnable functions that map $|vu|$ to the coefficients. $V_v$ defined in (42) can be computed similarly. Note that (44) does not require the computation of Clebsch-Gordan coefficients, the spherical harmonic functions, and all computations are in the matrix form where no for-loop is needed, so it is much faster than the computations in (42).

## H    MORE DETAILS OF SEC. 6

### H.1    A REMARK ON METRICS

When $N = 2$, *i.e.*, the input point clouds are $X_1$ and $X_2$, the metric we used is simply the averaged RRE/RTE of $(X_1, X_2)$ and $(X_2, X_1)$. Other popular metrics used in registration papers (Huang et al., 2021), like FMR and IR, are not suitable, because they measure the quality of the estimated correspondences, which our method does not compute. In addition, we do not use RR, because it is similar to the RRE and RTE metric (mean value v.s. threshold percentage), and it depends on a manually selected threshold for indoor scene, *i.e.*, 5 degrees and 2cm. This metric does not apply to other dataset than 3DMatch, e.g., the translation threshold is way too strict for outdoor dataset like KITTI, and is not applicable for dataset with unknown scale like BB.

## H.2 AN VERIFICATION OF EQUIVARIANCES

This subsection directly checks the relatedness of the learned vector field $v$. To verify the $SO(3)$-relatedness, *i.e.*, $v_X(rg) = rv_X(g)$, we compute loss $L_R = ||v_X(rg) - rv_X(g)||_F$ where $r$ is a random rotation, and $g$ is a random rigid transformation. This error will be close to zero if the equivariance holds.

On the other hand, we can also verify the $\sigma$-relatedness directly by computing $L_P = ||v_{\sigma X}(\sigma g) - \sigma v_X(g)||_F$, where $\sigma$ is a random permutation and $g$ is a random rigid transformation. Similarly, this error will be close to zero if the equivariance holds.

We compute $L_P$ and $L_R$ 3 times and present the mean results in Table 6. We observe that both errors are close to 0 with only floating point errors, suggesting these two equivariances hold.

| $L_R$ | $L_P$ |
|---|---|
| $1.76 \times 10^{-7}$ | $1.1 \times 10^{-7}$ |

Table 6: Rotation and permutation equivariance error.

## H.3 MORE RESULTS ON 3DMATCH, BB AND KITTI

We present more details of Eda on 3DL in Fig. 5. We observe that the vector field is is gradually learned during training, *i.e.*, the training error converges. On the test set, RK4 outperforms the RK1, and they both benefit from more time steps, especially for rotation errors.

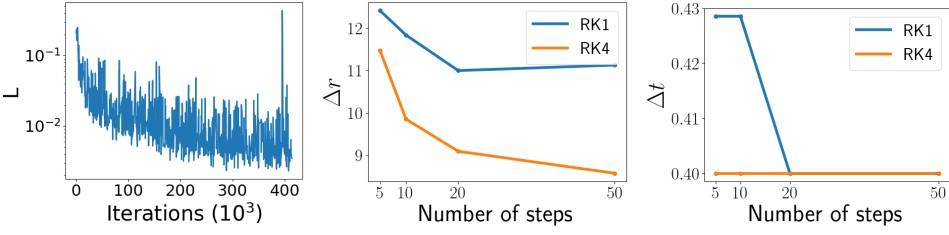

Figure 5: More details of Eda on 3DL. Left: the training curve. Middle and right: the influence of RK4/RK1 and the number of time steps on $\Delta r$ and $\Delta t$.

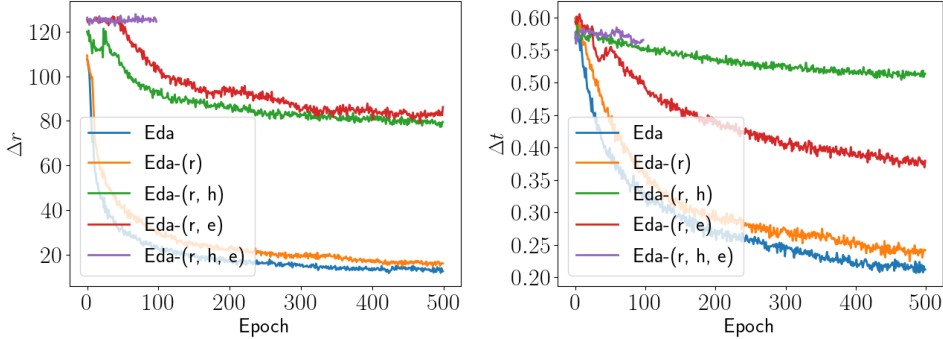

Figure 6: Validation error curves of all methods in Tab. 4. The training of Eda-$(r, h, e)$ is unstable and produces NaN value at the early stage.

We now provide more details for the ablation study reported in Tab. 4. The curve of validation errors of all methods are presented in Fig. 6. All methods use (RK1, 10) for sampling. Eda-$(r)$ satisfies all equivariances. Eda-$(r, h)$ breaks the first and third part of Prop. 4.7. Eda-$(r, e)$ and Eda-$(r, h, e)$ further break the second part of Prop. 4.6. The non-equivariant network is obtained by replacing the matrix (45) by a linear transformation with exactly the same number of parameters. All methods considered in this study contain exactly the same number of trainable parameters.

We provide the complete version of Table 2 in Table 7, where we additionally report the standard deviations of Eda.

Table 7: The complete version of Table 2 with stds of Eda reported in bracked.

|  | 3DM | | 3DL | | 3DZ | |
|---|---|---|---|---|---|---|
|  | $\Delta r$ | $\Delta t$ | $\Delta r$ | $\Delta t$ | $\Delta r$ | $\Delta t$ |
| FGR | 69.5 | 0.6 | 117.3 | 1.3 | – | – |
| GEO | 7.43 | 0.19 | 28.38 | 0.69 | – | – |
| ROI (500) | 5.64 | 0.15 | 21.94 | 0.53 | – | – |
| ROI (5000) | 5.44 | 0.15 | 22.17 | 0.53 | – | – |
| AMR | 5.0 | **0.13** | 20.5 | 0.53 | – | – |
| Eda (RK4, 50) | **2.38** (0.16) | 0.16 (0.01) | **8.57** (0.08) | **0.4** (0.0) | 78.74 (0.6) | 0.96 (0.01) |

We provide some qualitative results on BB datasets in Fig. 7. Eda can generally recover the shape of the objects.

A complete version of Tab. 3 is provided in Tab. 8, where we additionally report the standard deviations of Eda.

Table 8: The complete version of Table 3 with stds of Eda reported in brackets.

|  | $\Delta r$ | $\Delta t$ | Time (min) |
|---|---|---|---|
| GLO | 126.3 | 0.3 | 0.9 |
| DGL | 125.8 | 0.3 | 0.9 |
| LEV | 125.9 | 0.3 | 8.1 |
| Eda (RK1, 10) | 80.64 | 0.16 | 19.4 |
| Eda (RK4, 10) | 79.2 (0.58) | 0.16 (0.0) | 76.9 |

We provide a few examples of the reconstructed road views in Fig. 8.

# I   LIMIATION AND FUTURE WORKS

Eda in its current form has several limitations. First, Eda is slow when using a high order RK solver with a large number of steps. Besides its iterative nature, another cause is the lack of CUDA kernel level optimization like FlashAttention (Dao et al., 2022) for equivariant attention layers. We expect to see acceleration in the future when such optimization is available. Second, Eda always uses all input pieces, which is not suitable for applications like archeology reconstruction, where the input data may contain pieces from unrelated objects. Finally, in the future research, we plan to make Eda a foundation model by scaling up the training, so that it can handle different types of data and achieve higher precision. In particular, the scaling law (Kaplan et al., 2020) of Eda worths investigation, where we expect to see that an increase in model/data size leads to an increase in performance similar to image generation applications (Peebles and Xie, 2023).

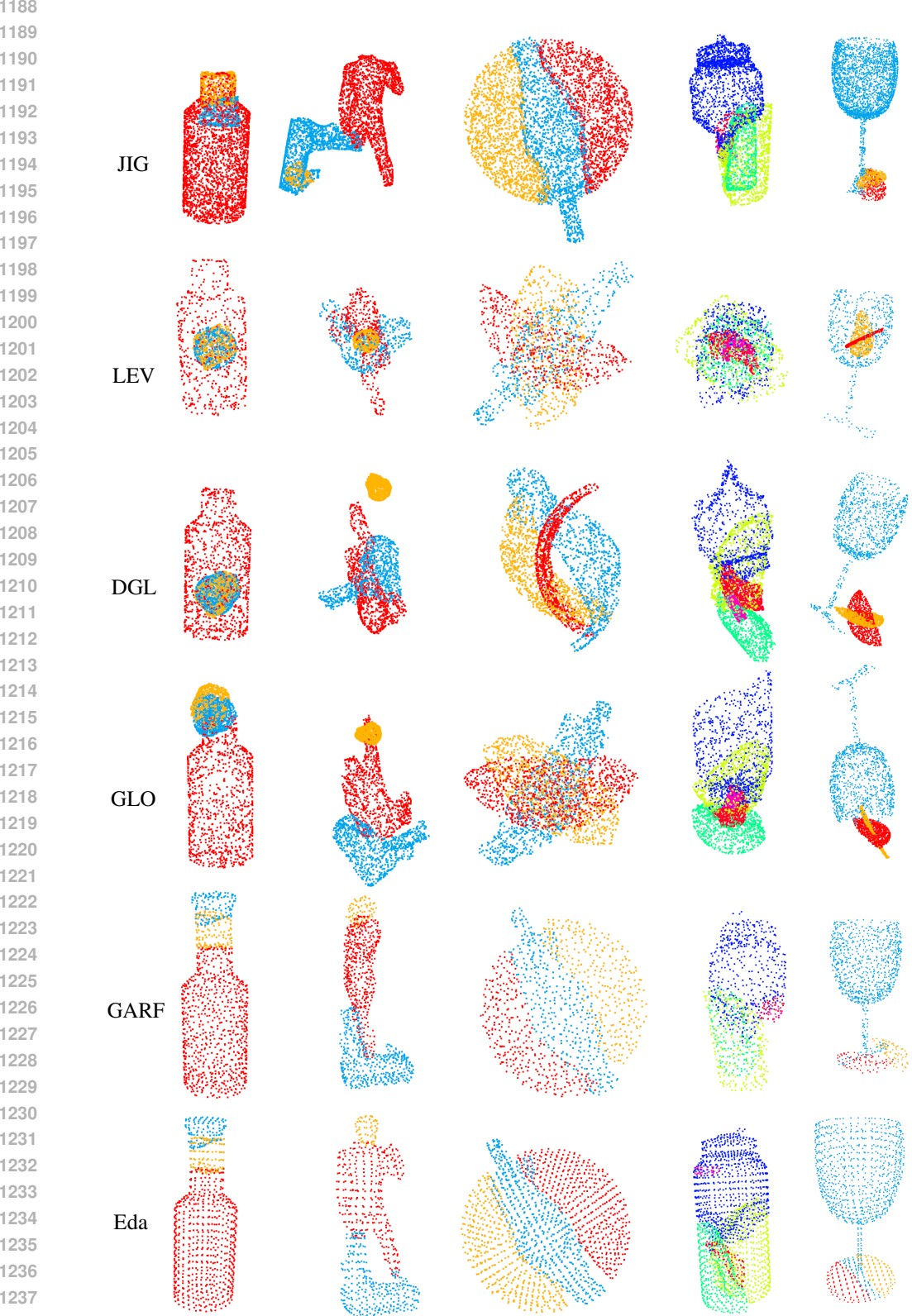

Figure 7: Qualitative results on BB. Zoom in to see details.

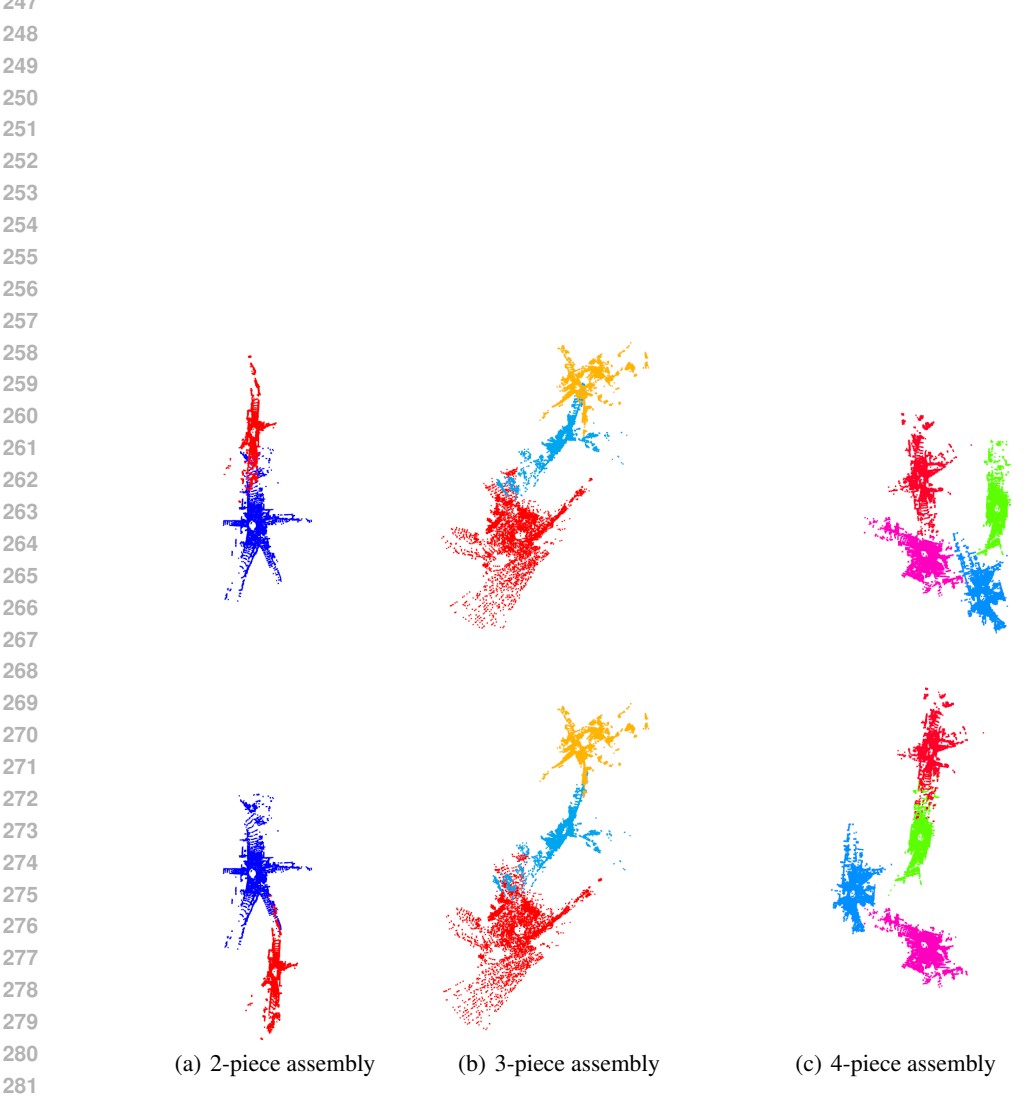

(a) 2-piece assembly  (b) 3-piece assembly  (c) 4-piece assembly

Figure 8: Qualitative results of Eda on kitti. We present the results of Eda (1-st row) and the ground truth (2-nd row). For each assembly, Eda correctly places the input road views on the same plane.

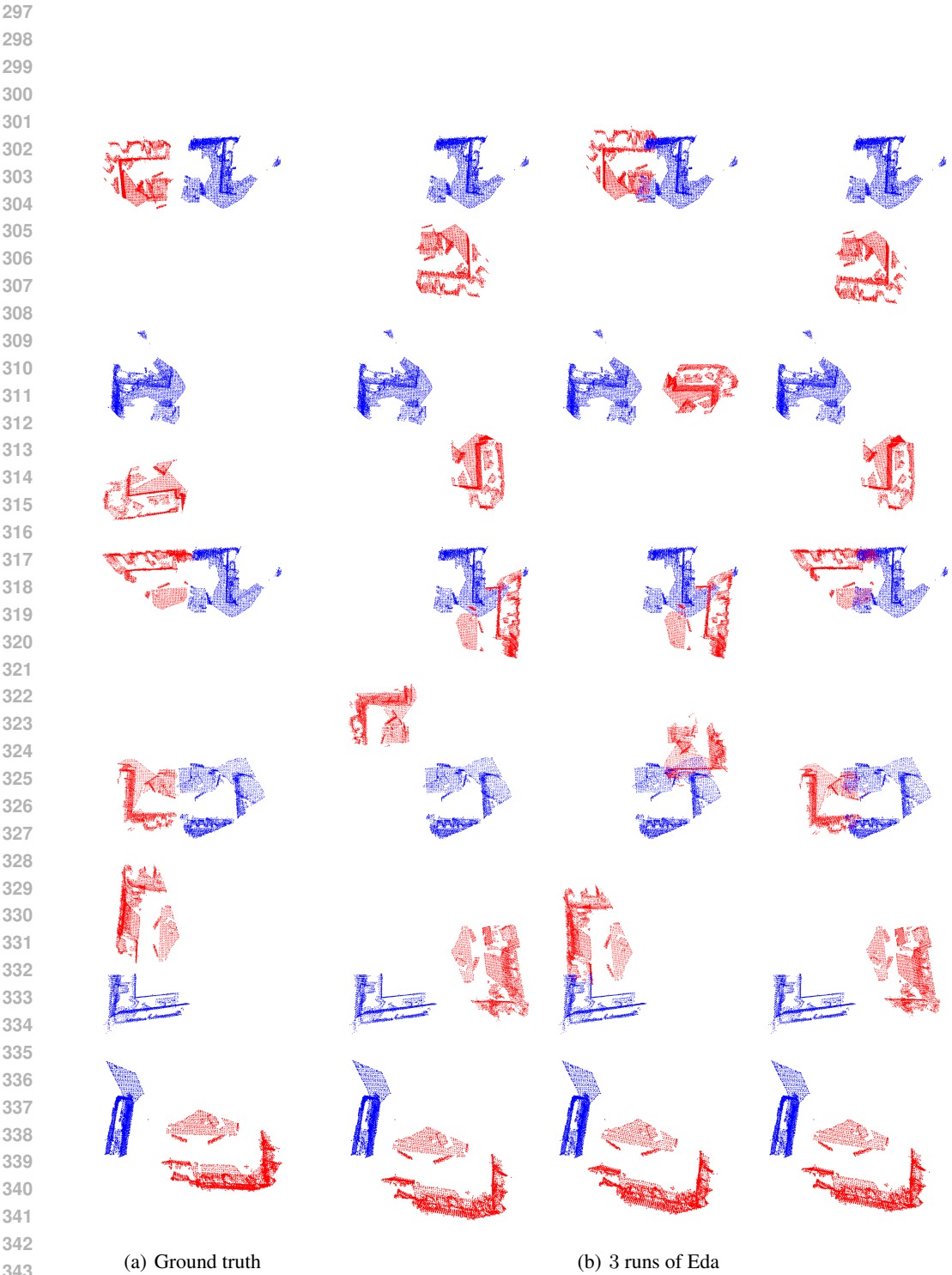

(a) Ground truth          (b) 3 runs of Eda

Figure 9: Qualitative results of Eda on 3DZ. Cameras are set to look at the room from above.

