# OpenReview forum: "Equivariant Flow Matching for Point Cloud Assembly"
_ICLR.cc/2026/Conference — Submitted to ICLR 2026_

### Official Review · Reviewer_zRLa · 2025-10-31

**Soundness:** 3
**Presentation:** 3
**Contribution:** 3
**Rating:** 8
**Confidence:** 3

**Summary:**

This work proposes an equivariant flow matching framework for multi-piece point cloud assembly tasks. The key idea is to employ a vector field parameterized by equivariant networks on an invariant base distribution to ensure the output distribution is equivariant to SO(3) rotations and permutations. Additionally, the training efficiency is enhanced by considering modified samples and random noises with minimum distance across all possible rotations. Overall, the experimental results show that the proposed framework achieves better results than existing baselines, and the ablation studies well validate the effectiveness of the proposed network.

**Strengths:**

- This work proposes a clear formulation of the base distribution and vector field network assumptions to ensure the output distribution is equivariant to the group. The formulation of the base distribution ($\left(U_{\mathrm{SO}(3)} \otimes \mathcal{N}(0, \omega I)\right)^N$) and equivariant layers appear to be well-suited for these assumptions.
- The experimental results demonstrate strong performance of the proposed framework, achieving better results in both pair-wise registration and multi-piece assembly.
- Additionally, the manuscript validates the proposed components through the ablation study in Table 4, which further justifies the need for rotation correction and equivariant networks.

**Weaknesses:**

There is only one minor concern in the current manuscript:
Regarding the ablation study, it is mentioned that the proposed equivariant network is replaced with a non-equivariant counterpart. It would be beneficial to provide more details on the non-equivariant network and describe the exact changes in the Appendix to ensure the experiment's fairness.

**Questions:**

See the weakness section.

---

> ### Author Response · Authors · 2025-11-18
>
> We thank the reviewer for the positive feedback. We address the concerns below.
>
> 1. **It would be beneficial to provide more details on the non-equivariant network and describe the exact changes in the Appendix to ensure the experiment's fairness.**
>
>
> We break the equivariance of the attention layer by breaking the form of matrix in Eqn. (45). We simply replace this matrix by a linear layer with the same number of parameters. This was presented in line 1076: "The non-equivariant network is obtained by replacing the matrix (45) by a linear transformation with exactly the same number of parameters"

---

> > ### Comment · Reviewer_zRLa · 2025-11-28
> >
> > Thanks for your responses, and they have addressed my concern. I will maintain the current rating.

---

### Official Review · Reviewer_QEHc · 2025-11-01

**Soundness:** 3
**Presentation:** 2
**Contribution:** 2
**Rating:** 4
**Confidence:** 4

**Summary:**

This paper proposes Eda (Equivariant Diffusion Assembly), a novel correspondence-free, multi-piece point cloud assembly model built on equivariant flow matching. The key theoretical contribution is to show that learning equivariant distributions can be reduced to learning related vector fields, provided the initial noise distribution is invariant. Building on this, the authors design an SE(3)^N-equivariant flow-matching framework where the equivariance of the learned distribution is guaranteed by construction. Eda parametrizes vector fields through an equivariant neural network and introduces an equivariant path construction that improves data efficiency during training. Experiments on 3DMatch, BB, and KITTI demonstrate strong quantitative improvements over state-of-the-art baselines, including robust performance even on non-overlapping fragments (3DZeroMatch). The results support both theoretical soundness and empirical effectiveness.

**Strengths:**

- The paper provides a solid theoretical framework by reducing equivariant distribution learning to learning related vector fields. The derivations appear rigorous and consistent, although I did not verify every step in detail.

- The use of E3NN-based equivariant attention and Croco blocks makes the approach practical.

**Weaknesses:**

- Although the paper provides theoretical guarantees for SO(3)^N-equivariance, the empirical validation is mostly indirect. The ablation studies show performance drops when removing the equivariant backbone or path, suggesting that equivariance helps, but this only demonstrates effectiveness, not faithful equivariant behavior. A more rigorous validation would involve explicitly applying controlled rotations to input fragments (right-multiplication in SO(3)^N), or global rotations (left-multiplication), and verifying whether the predicted poses transform accordingly. Such experiments would directly confirm that the learned flow v_X(g) satisfies the claimed equivariance relation v_{rX}(rg)=r\,v_X(g).

- The authors claim that their model achieves permutation equivariance; however, no direct experiment is provided to validate this claim. It remains unclear how the predicted poses change when the input order of point clouds is permuted.

- Weak Experimental Validation. The experimental section is relatively weak and limited in scope. The first experiment focuses on pairwise registration, which does not align with the paper’s stated goal of multi-piece point cloud assembly. The comparison set is also narrow and excludes strong, widely recognized metrics commonly adopted in the pairwise registration literature and also missing many prevalent pairwise methods such as FCGF, Predator and BUFFER. The multi-piece assembly evaluation is further constrained, only 2–8 fragments on synthetic datasets, which makes it difficult to assess the method’s scalability or robustness in realistic settings. Fig. 4 further indicates limited generalization capacity, performance degrades notably on unseen fragment lengths, revealing the model’s fragility. Including results on more comprehensive datasets such as Fantastic Breaks or FRACTURA would significantly strengthen the empirical claims.
Finally, the KITTI experiment appears loosely connected to the main task, and its relevance to point cloud assembly is not clearly justified, leaving the overall empirical validation unconvincing.

Overall, the experimental validation is quite weak and does not convincingly demonstrate the real effectiveness of the proposed Equivariant Flow Matching. The experiments are limited in scope, and key claims, such as equivariance and invariance, are only indirectly supported. These issues collectively make me lean toward rejecting this paper at its current stage.

**Questions:**

See weaknesses

---

> ### Author Response · Authors · 2025-11-18
>
> We thank the reviewer for the time and effort. We address the concerns below.
>
> 1. **Q1, Q2. The verification of rotation  and permutation equivariance.**
>
> Thanks for your suggestion.
>
> As for the rotation equivariance, it is difficult to check the equivariance by checking the predicted poses, because the prediction is non-deterministic. Instead, to verify the relatedness $v_{X}(rg)=r v_X(g)$, we can directly measure $L_{R}=||v_{X}(rg) -  rv_X(g)||_F$ where $r$ is a random rotation and $g$ is a random rigid transformation. This error will be close to zero if the equivariance holds.
>
> On the other hand, the permutation equivariance is simply the result that the order of inputs is not used (all computation remains the same regardless of the order).  We can also verify this directly by computing $L_{P} =||v_{\sigma X}( \sigma g) - \sigma v_X(g)||_F$, where $\sigma$ is a random permutation and $g$ is a random rigid transformation.
>
> We have now added the following table in Appx. H.2 in our revised manuscript, where we observe that both errors are close to $0$ with only floating point errors, suggesting both equivariances hold.
>
> | $L_{R}$ | $L_{P}$ |
> | --- | --- |
> | $1.76 \times 10^{-7}$   | $2.51\times 10^{-7}$   |
>
>
> 2. **The first experiment focuses on pairwise registration, which does not align with the paper’s stated goal of multi-piece point cloud assembly..... excludes strong, widely recognized metrics ....FCGF, Predator and BUFFER.... only 2–8 fragments on synthetic datasets,.....Fig. 4 further indicates limited generalization capacity, performance degrades notably on unseen fragment lengths..... datasets such as Fantastic Breaks or FRACTURA would significantly strengthen the empirical claims. ...Finally, the KITTI experiment appears loosely connected to the main task, and its relevance to point cloud assembly is not clearly justified**
>
>
> **The alignment with the goal** Our method focus on the $N \geq 2$ pieces problem, so the pair-wise setting ($N=2$) clearly aligns with our goal as a special case.
>
> **Metric** The metric we used is simply RRE and RTE when $N=2$, which we believe is one of the "strong, widely recognized metrics". Note that this metric was also used in the Predator paper. Other metrics used in that paper, for example FMR and IR, are not applicable because they measure the quality of the estimated correspondences, which our method does not compute. In addition, we do not use RR because it is too similar to the RRE and RTE metric (mean value v.s. threshold percentage), and it depends on scales, ie, it has a manually selected translation error threshold 2cm, which is way too strict for outdoor dataset like KITTI, and is not applicable for dataset with unknown scale like BB. We have now added this clarification to Appx. H. 1 in our revised manuscript.
>
> **Baseline methods** We have considered the state of the art methods ROI (2023)  GEO (2022)  AMR (2025) as the baselines, which have been shown to be stronger than FCGF (2019) Predator (2021) BUFFER (2023) in previous studies. See the AMR paper for example.
>
> **The datasets**  The BB dataset is the standard large-scale dataset of pieces assembly. Please check the papers of all the baseline methods. We select 2-8 pieces simply because that constitutes the majority of the data (70\%) and helps to avoid some ill-posed instances, e.g, pieces that are too small to be visible. See Fig. 3 in [1] for an example of those pieces. In addition,
>
> 1). **Fantastic Breaks** is an impainting dataset, we can convert it into a **two-piece** assembly dataset ($N=2$), which does not satisfy our requirement that some samples need to have $N >2$ pieces in this experiment. We do not know any way to make it a assembly dataset with $N > 2$.
>
> 2). **FRACTURA** is a really small dataset containing **only $27$ real samples** from **$1$ class**, where each sample contains **at most 4 pieces**. (it also contains about $300$ synthesized samples, which are synthesized exactly the same way as BB).
>
> In summary, in terms of scalability and diversity of the dataset, we believe BB (with more than 2k samples and about 10 classes) is a better option.
>
>
> **KITTI**  We assemble the KITTI data to reconstruct street scenes, which we believe aligns well with the goal of the paper.
>     As for the ablation of piece number, the results show a dependency on the training and test sequence lengths.
>     Such a dependency is well known in transformer-type models, see [2], and we clarify that this also holds for our model.
>     This is not a surprising result, as our model is also based on a Transformer.
>     It is unfair to claim this is just a limitation of our method's capacity.
>
> [1] Fragmentdiff: A diffusion model for fractured object assembly. In SIGGRAPH Asia 2024 Conference Papers, pages 1–12, 2024
>
> [2] Press, O., Smith, N. A., \& Lewis, M. (2021). Train short, test long: Attention with linear biases enables input length extrapolation. arXiv preprint arXiv:2108.12409.

---

> > ### Comment · Reviewer_QEHc · 2025-11-28
> > **Comments**
> >
> > Thank you for the rebuttal and for the clarifications. However, several essential concerns remain unresolved, and therefore it is still difficult for me to raise my score at this stage.
> >
> > (i) I remain unconvinced about the necessity and motivation for developing equivariant flow matching. Unlike transformation-invariant properties that enhance the discriminative power of geometric features, equivariance only prescribes how representations transform under rotations, and this alone does not directly explain why it should improve inference precision. At present, there is no concrete empirical evidence showing that enforcing equivariance provides measurable benefits for the assembly task. Furthermore, the practical significance of introducing equivariance into this framework remains unclear. A more rigorous justification or ablation is required to support this design choice.
> >
> > (ii) It is also unclear how the permutation and rotation equivariant properties are fundamentally integrated into the flow-matching pipeline. In the reviewer's understanding, both rotation and permutation equivariance can be achieved purely through architectural design (e.g., vector-neuron–based models), and it is not evident why they need to be embedded into the flow-matching framework. After training the score function within your flow-matching setup, how does the system behave at inference time? If the fragments are randomly rotated and permuted, does the model still require running the full flow-matching denoising process to recover poses? This seems inconsistent with the usual interpretation of rotation equivariance, where feature-level transformations should be sufficient without requiring re-inference of the full flow.
> >
> > (iii) The experimental design also deviates from what readers naturally expect for a point cloud assembly task. While pairwise alignment may be viewed as a sub-problem, readers reasonably expect thorough multi-piece evaluations. However, the multi-piece experiments are extremely limited: the authors only selected a small number of 2–8 piece cases from the everyday subset of the BB dataset. This suggests that the actual number of evaluated samples is very limited, and that the method has not been sufficiently tested on the BB dataset as a whole.
> >
> > (iv) The rationale for excluding real fractured datasets such as Fantastic Breaks or FRACTURA is also unconvincing. Unlike the CAD-synthesized fragments in BB, which possess artificially clean and idealized separation surfaces, these real-world fragments exhibit genuinely fractured, crack-like, irregular break surfaces that are fundamentally absent in CAD-generated data. Such real fracture geometry is crucial for assessing real-world assembly performance. Moreover, the rebuttal argues that these datasets contain only N=2 fragments and therefore do not fit the experimental protocol, yet the paper itself includes extensive two-piece experiments on 3DMatch and 3DLoMatch. This inconsistency makes it difficult for the reviewer to accept the explanation.
> >
> > (v) The authors are also encouraged to provide clearer specifications of the backbone architecture used for fragment encoding, including architectural components, configurations, and the exact downsampling strategy employed. These technical details are essential for a proper assessment of the method and for ensuring reproducibility.

---

> > > ### Author Response · Authors · 2025-11-28
> > >
> > > Thanks for the reply.
> > >
> > > It seems that the reviewer just repeated the previous concern and added some new concerns like "what is it?", "how does it work?". We suggest the the reviewer read our reply before replying to us, and we believe that the added preliminary questions should be asked in the first place, not at the end of the disuccsion period. Nevertheless, we will address the concerns below.
> > >
> > > We kindly ask the reviewer to be open minded, and do not reject a paper simply because you are not familiar with the equivariance theory.
> > >
> > > ----
> > > 1. **..unconvinced about the necessity and motivation for developing equivariant flow matching...  concrete empirical evidence showing that enforcing equivariance provides measurable benefits for the assembly task**
> > >
> > > See ablation in table 4 for an "emperical evidence". As for the motivation, it was clearly stated in the introduction (line 33): the equivariance is a basic law underlying the assembly task, the use of equivariances provides extra guidance of the assembly.
> > >
> > > 2.  **"why they (equivariances) need to be embedded into the flow-matching framewor"** and how does it work?
> > >
> > > The rationale is in Th.4.2, Cor 4.4 and Prop.4.6, which explains how the equivariances of conditional distributions lead to related vector fields and then lead to equivariant networks. This is consistent with "the usual interpretation of rotation equivariance" in terms of related vector fields. More intuition about "how does equivariances work" in inference with N=2 for a similar denoise model can be found in Ryu et al., 2024.
> > >
> > >  3.4. **The experiments. "This suggests that the actual number of evaluated samples is very limited..." "which possess artificially clean and idealized separation surfaces" "Moreover, the rebuttal argues that these datasets contain only N=2 fragments and therefore do not fit the experimental protocol"**
> > >
> > >
> > > We have already replied to a part of this concern in our last reply. Now we just quote it:
> > >
> > > "We select 2-8 pieces simply because that constitutes the majority of the data (70%) and helps to avoid some ill-posed instances,e.g, pieces that are too small to be visible." "In summary, in terms of scalability and diversity of the dataset, we believe BB (with more than 2k samples and about 10 classes) is a better option." (than any dataset you mentioned)
> > >
> > > In addition, the statement is not true for the BB dataset. BB is a physics simulated dataset, which does not contain any "clean and idealized separation surfaces", instead, they contain "crack-like, irregular break surfaces". In addition, since we already have N=2 pieces experiments using 3DM/3DL/3DZ, which are real datasets, why is it necessary to consider N=2 again using an impainting dataset which is not designed for assembly? On the other hand, FRACTURA is limited in terms of scale and diversity, not the piece number N. That was also stated in our previous reply.
> > >
> > >
> > >
> > >  5. **..clearer specifications of the backbone architecture....**
> > >
> > >  All details were already clearly stated in Sec.6 and Appx.G.

---

### Official Review · Reviewer_xdq7 · 2025-11-02

**Soundness:** 3
**Presentation:** 2
**Contribution:** 3
**Rating:** 6
**Confidence:** 3

**Summary:**

The paper proposes Eda, an equivariant flow-matching framework for assembling 3D point cloud fragments. It combines E(3)-equivariant layers with a flow-based architecture to enable efficient SE(3)-equivariance learning . Experiments on 3DMatch, 3DLoMatch, and Breaking Bad show over 50% lower rotation error than baselines. While theoretically elegant and empirically strong, the paper lacks ablation on equivariance, efficiency analysis, and tests on noisy real-world point clouds.

**Strengths:**

- The paper provides a solid theoretical foundation for framing point cloud assembly as a flow matching problem.
- On 3DMatch and 3DLoMatch, Eda achieves >50% lower rotation errors than GEO/ROI/AMR baselines. It also handles non-overlapping fragments (3DZeroMatch) where correspondence-based methods fail entirely.
- The paper provides a good ablation study on varying different settings.

**Weaknesses:**

- How does the method work on an untrained category of assembly?
- How does the method work if equivariance is ablated? The author might want to consider comparing with the same architecture but only lack of equivariance for paper completeness.
- While the theory side is useful, a better native like intuitive diagram would help aid readability of the paper.
- While the paper claims that learning related vector fields provides a more efficient alternative to full equivariant flow modeling, the evidence remains largely qualitative. The only quantitative indicator is a reduction in assembly runtime (≈ 19 minutes per object versus ≈ 34 minutes for diffusion-based baselines). A more detailed analysis of the computational efficiency would benefit the paper’s completeness (e.g. training convergence, FLOPs, memory footprint, scalability curve, etc.)

**Questions:**

- How does the method generalize to real world noisy point clouds? - As one of the benefits of flow base method is its generalization ability to messy real world applications.

---

> ### Author Response · Authors · 2025-11-18
>
> We thank the reviewer for the time and effort. We address the concerns below.
>
>
> 1. **How does the method work on an untrained category of assembly?**
>
> In its current form, our model works on the untrained test class, as long as it belongs to the same coarse class as the training data, i.e. training and test data is not drastically different. For example, on 3DMatch (the "indoor" dataset), the method works on the test data of 7-scene subset (containing the "kitchen" class), while the training set contains different classes like "stairs" and "office".
>
> In the future work, we plan to make our model a foundation model by scaling up the training, so that it can handle very different data. We have now added this remark in our revised manuscript (line 1171)
>
> 2. **How does the method work if equivariance is ablated? The author might want to consider comparing with the same architecture but only lack of equivariance for paper completeness.**
>
> In that case the model still works, but not as effective, and Yes! Please see row Eda-(r, e) in Tab.4, where we replaced the equivariant component with a non-equivariant one with the same number of trainable parameters. The training curve of this ablation can be found in Fig. 6.
>
> 3. **While the theory side is useful, a better native like intuitive diagram would help aid readability of the paper.**
>
> Thanks for the suggestion. However, we found it hard to explain the algebra structures using diagrams, so we provided a walk through of the whole theory in Appx C, explaining the theory in plain language with no terminology. We mentioned this section at the end of the introduction. We hope this can make the paper easier to read.
>
>
> 4. **While the paper claims that learning related vector fields provides a more efficient alternative to full equivariant flow modeling, the evidence remains largely qualitative. The only quantitative indicator is a reduction in assembly runtime (= 19 minutes per object versus = 34 minutes for diffusion-based baselines). A more detailed analysis of the computational efficiency would benefit the paper’s completeness (e.g. training convergence, FLOPs, memory footprint, scalability curve, etc.)**
>
> As we mentioned in line 38, the vector field learning is more efficient than the full equivariant model like [1] ([1] is not a diffusion model). This is because to assemble $N$ pieces, the related field method (like ours)  needs a SO(3)-equivariant network (which does not depend on $N$). While a complete equivariant model would require a $SO(3)^N$-equivariant network (which depends on $N$). Note that the memory usage of a $SO(3)^N$-equivariant network like [1] grows exponentially with $N$. That is one reason why a full equivariant model like [1] is only practical when $N$ is small, i.e, $N=2$.
>
> In practice. For $N=2$, [1] requires about 2G GPU memory per sample during training, while our method only requires 0.6G.
> For larger $N$, it is hard to compare these two types of models because we do not know of any practical full equivariant model for $N > 2$.
>
>
> [1] Ziming Wang and Rebecka Jörnsten. Se (3)-bi-equivariant transformers for point cloud assembly. In The Thirty-eighth Annual Conference on Neural Information Processing Systems (NeurIPS), 2024.
>
> 4. **How does the method generalize to real world noisy point clouds? - As one of the benefits of flow base method is its generalization ability to messy real world applications.**
>
> Both the 3DM and KITTI are real datasets, and they contain noise of different magnitude.

---

### Official Review · Reviewer_9ojc · 2025-11-07

**Soundness:** 2
**Presentation:** 2
**Contribution:** 2
**Rating:** 2
**Confidence:** 4

**Summary:**

This paper introduces an equivariant solver for assembly tasks based on flow matching. Theoretically, the authors demonstrate that learning equivariant distributions via flow matching requires learning corresponding equivariant vector fields. Building upon this result, this paper proposes the Equivariant Diffusion Assembly (EDA) model, which learns these vector fields conditioned on the input pieces. Furthermore, they construct an equivariant sampling path for EDA, a design that ensures high data efficiency during training.

**Strengths:**

1.	The motivation is clear.
2.	Mathematical descriptions are sufficient.

**Weaknesses:**

1. Inadequate literature review on key related works, especially patch-based registration and non-overlap registration methods.
2. Unsubstantiated claim of solving the multi-piece problem, as the method and experiments primarily focus on two-piece problem with one experiment for multi-piece problem on BB dataset.
3. Unclear novelty and contribution, as the method heavily builds upon established components without a clear clarification.
4. Incomplete experimental validation, due to an insufficient number of compared methods, limited datasets, and a lack of rigorous testing on multi-piece cases.

**Questions:**

1.	This paper claims to address the multi-piece assembly. There are lots of patch-based point cloud registration methods that have not been carefully discussed: [1] Zhao, T., Tian, T., Zou, X., Yan, L., & Zhong, S. (2025). Robust Point Cloud Registration via Patch Matching. IEEE Transactions on Geoscience and Remote Sensing. [2] Zhao, T., Li, L., Tian, T., Ma, J., & Tian, J. (2023). Patch-guided point matching for point cloud registration with low overlap. Pattern Recognition, 144, 109876. [3] Qin, Z., Yu, H., Wang, C., Guo, Y., Peng, Y., Ilic, S., ... & Xu, K. (2023). Geotransformer: Fast and robust point cloud registration with geometric transformer. IEEE Transactions on Pattern Analysis and Machine Intelligence, 45(8), 9806-9821.
2.	This paper is also targeted at non-overlap assembly, which is also not a new problem. For example, the following strengths and weaknesses should also be discussed: [1] Xu, J., Dai, H., Hu, X., Fan, S., & Ke, T. (2024). SCREAM: Scene rendering adversarial model for low-and-non overlap point cloud registration. IEEE Transactions on Geoscience and Remote Sensing. [2] Xu, J., Zhang, Y., Zou, Y., & Liu, P. X. (2023). Point cloud registration with zero overlap rate and negative overlap rate. IEEE Robotics and Automation Letters, 8(10), 6643-6650.
3.	This method seems to be constructed on the basis of Ryu et al. (2024). The main difference between this method and Ryu et al. lies in the Brownian diffusion on SO(3), while the method proposed by Ryu et al. solves the more general multi-piece problem.
4.	The related work is not organized very well; readers can not catch the differences between this paper and existing methods. It is recommended to separate this section into several related subsections.
5.	The preliminaries section is too long; it is suggested to shorten it and only put the important information into the main text.
6.	Although this paper claims the use of multiple pieces, the definition and experiments mainly focus on a two-piece problem with one dataset to validate the multiple pieces problem.
7.	SO(3)-equivariant networks are also widely used in 3D. The introduction of this network in the main text needs to be shortened.
8.	Moreover, the vector field in flow matching is also a well-known definition, whose introduction also needs to be shortened.
9.	If all experiments are conducted on only one multi-piece problem, then it is not very suitable to claim that solving a multiple-piece problem.
10.	Equivariant flow is widely used in 2D computer vision and 3D molecular generation. It is hard to find the main contribution when compared with existing equivariant flows. It is recommended to re-emphasize your contribution in Section 4.2.
11.	Sampling with the RUNGE-KUTTA method is also not a novel technique in flow matching.
12.	In the implementation, the vanilla Transformer is employed. Why not use a point transformer with permutation-equivariant or a transformer with SO(3)equivariant in recent years? They might obtain better performance. Most importantly, there are lots of innovations in point cloud process. Building your methods on existing practices will be better.
13.	There are also existing blocks employed in your architecture. The main contributions can not be clearly understood. Moreover, it is not recommended to rename self-attention and cross-attention with a new name croco block. They can be easier to understand than giving a new name,
14.	As for experiments, there are lots of pages for introducing existing details. Therefore, fewer pages are left for experiments, which leads to incomplete experimental validation.
15.	In Figure 3, if the 8-piece assembly process is to be displayed, 8 different colors should also be given. It is highly suggested to validate your method on multiple piece problems.
16.	Moreover, the compared methods in point cloud registration are not comprehensive enough. It is a widely studied area, and more up-to-date methods should be compared.
17.	Importantly, only two datasets are limited. There are also many datasets related to point cloud registration.

---

> ### Author Response · Authors · 2025-11-18
>
> Dear reviewer 9ojc,
>
> Thanks for your time reviewing our work.
>
> We understand that this paper is theoretical, and some background in equivariant networks is needed to understand the paper. We kindly ask you to be open minded, and do not reject the paper simply because you do not understand it.
>
> We tried our best to address your concerns below. However, if our reply fails to provide new insights of the work, please let us know, or at least **lower your confidence level**. The current confidence 4 means "It is unlikely... that you did not understand some parts of the submission", which we do not believe is an accurate reflection of the current situation, and would cause difficulties for us and the AC to correctly interpret your reviews.
>
> We thank you again for reading (at least some parts of) our paper.
>
> Best
>
> Authors
>
> ------
> 1. **(Q3. Q6)  Eda only works for 2-piece problem.  ... This method seems to be constructed on the basis of Ryu et al. (2024)...the method proposed by Ryu et al. solves the more general multi-piece problem... the definition and experiments mainly focus on a two-piece problem..**
>
> This is not true for either our method or Ryu's. Ryu's method is a two-piece method, and our method is a multi-piece method. This was clearly stated in line 79. Our whole paper is developed under the multi-piece setting, from the definition to the theory and experiments. The definition is also quite clear: $N \geq 2$ pieces are considered in our work. See line 101. This definition is used through out the theory. As for the experiments, the BB and KITTI datasets we used are multi-piece datasets. See our reply 4 for datasets.
>
> 2. **(Q1, Q2) Discussion of the  "patch based" and "non-overlap" methods.**
>
> These types of methods are already included in our discussion.  We do not see any necessity to discuss the papers you mentioned separately (excluding GEO, which was already considered as a baseline method.)
>
>     1). The "patch-based" methods belong to the correspondence-based methods. See Appx. B for discussion. In particular, the Geotransformer paper is included there, and is also used as the baseline in Sec 6.2.
>     2). The non-overlap assembly papers you mentioned belong to two-piece non-deterministic methods. See Tab. 5.
>
> 3. **(Q4, Q5, Q7, Q8, Q10, Q14) : The paragraphs ( related work, preliminaries, introduction of equivariant networks and vector fields, Sec.4.2,  introducing existing details ) are too long/ too short/ not clear/need to be "re-emphasize ".**
>
> We do not see any necessity to make any modification to those sections based on the reviewer’s comment.
>
>     1). The related work section is already well-organized, with each paragraph covering one aspect of connections.
>     2). We will not shorten the preliminaries, including the equivariant network and vector field part, because that information is necessary for readers to understand the theory, and the notations established there are used in later parts of the work.
>     3). There is no need to "re-emphasize" Sec. 4.2, because all theory developed there is our contribution. The connection with existing equivariant flow methods was clearly stated in line 81.
>     4). What exactly are the "pages for introducing existing details" before the experiments section? Deleting any part of the paper will break the structure of the work. We will keep it as it is.
>
> 4. **(Q16, Q17). ....more up-to-date methods should be compared.... only two datasets are limited....**
>
> See our reply 3 to reviewer QEHc for some discussions of baseline methods and datasets.
> We would also like to remind the reviewer that we also tested Eda on the KITTI dataset, which is a multi-piece assembly dataset.
>
> Also, if the reviewer feels more methods/datasets should be included, please be more explicit which one should be included, instead of using vague expression like "more methods", "only two datasets are limited".
>
>
>
> 5. **(Q9) If all experiments are conducted on only one multi-piece problem, then it is not very suitable to claim that solving a multiple-piece problem.**
>
> See the above reply for datasets. In addition, Eda solves the multi-piece problem. See reply 1.
>
> 6. **(Q11) Sampling with the RUNGE-KUTTA method is also not a novel technique in flow matching.**
>
> We never claimed that to be our contribution.
>
> 7. **(Q12, Q13)...You use vanilla transformer.. and what is Croco? They can be easier to understand than giving a new name,**
>
> This is not true. We did not use vanilla transformers. Instead, we built the network on E3NN and Croco. This was clearly stated in Sec 6.
>
> Croco is a classic architecture. Even if you are not familiar with it, you can find the citation in Sec. 6: Croco (Weinzaepfel et al., 2022).
>
>
> 8. **(Q15)...8 colors In Figure 3... It is highly suggested to validate your method on multiple piece problems.**
>
> There are 8 colors there. Zoom in to see the details.

---

> ### Comment · Reviewer_9ojc · 2025-11-27
>
> I have read the authors' response and have thus raised the rating to 4.

---

### Author Response · Authors · 2025-11-18
**Summary of changes**

We have made the following changes to our manuscript based on the reviews:
1. Added a remark on the metric we used in Appx.H.1.
2. Added a direct verification of the equivariances (relatedness) of the learned vector field in Appx.H.2.
3. Added a remark on scaling up the training in the future work (line 1171).

---

### Meta-Review · Area_Chair_Q3xw · 2025-12-17

**Summary:**

Reviewers found the theoretical formulation of equivariant flow matching elegant and the empirical results promising, but raised substantial concerns about clarity of contribution, experimental scope, and motivation. In particular, several reviewers questioned whether embedding equivariance into flow matching yields clear practical benefits and found the multi-piece assembly evaluation limited relative to the paper’s claims.

Although some concerns were partially addressed in the rebuttal, the remaining gaps in empirical validation and positioning led to an overall borderline-rejection.

**Reviewer Concerns:**

The rebuttal partially addressed concerns by (i) adding a direct numerical verification of rotation/permutation equivariance/relatedness of the learned vector field (Appx H.2), (ii) clarifying the registration metric choice (Appx H.1), and (iii) pointing to an explicit ablation where removing equivariance degrades performance (Tab. 4, incl. the non-equivariant replacement). It also resolved at least one reviewer’s misunderstanding about “two-piece only” and led to an upward score change (9ojc -> 4).

However, several concerns remain outstanding: the motivation/necessity of embedding equivariance specifically into flow matching was not convincingly argued for the skeptical reviewer (QEHc), and the empirical validation is still viewed as narrow for a “multi-piece assembly” claim (limited BB subset selection; limited stress-tests on broader piece counts / more realistic fractured datasets; and missing broader baselines in pairwise registration, per QEHc/9ojc). Reproducibility/detail concerns also linger (clearer backbone/encoding/downsampling details requested by QEHc), and some readers still find the paper’s narrative/presentation burdensome relative to the experimental scope.

**Reviewer Scores:**

Reviewer 9ojc explicitly updated their rating upward after reading the rebuttal (2 -> 4). Reviewer zRLa indicated their concern was addressed and would keep the current rating (8, moderate confidence & short and vague and not detailed comments -> therefore meta review downweighted its value)

Reviewer xdq7’s main questions (equivariance ablation and efficiency/equivariance verification) were directly answered with pointers to Tab. 4/Fig. 6, Appx H.2, and added efficiency discussion, so a small increase is plausible. Reviewer QEHc reiterated that key motivation/experimental-scope concerns remained unresolved after the rebuttal and did not signal any willingness to raise the score (4).

---

### Decision · Program_Chairs · 2026-01-26

Reject